# How to Reuse and Compose Knowledge for a Lifetime of Tasks: A Survey on Continual Learning and Functional Composition

**Jorge A. Mendez**                                    *jmendez@csail.mit.edu*
*Computer Science and Artificial Intelligence Laboratory*
*Massachusetts Institute of Technology*

**Eric Eaton**                                    *eeaton@seas.upenn.edu*
*Department of Computer and Information Science*
*University of Pennsylvania*

**Reviewed on OpenReview:** *https://openreview.net/forum?id=VynY6Bk03b*

## Abstract

A major goal of artificial intelligence (AI) is to create an agent capable of acquiring a general understanding of the world. Such an agent would require the ability to continually accumulate and build upon its knowledge as it encounters new experiences. Lifelong or continual learning addresses this setting, whereby an agent faces a continual stream of problems and must strive to capture the knowledge necessary for solving each new task it encounters. If the agent is capable of accumulating knowledge in some form of compositional representation, it could then selectively reuse and combine relevant pieces of knowledge to construct novel solutions. Despite the intuitive appeal of this simple idea, the literatures on lifelong learning and compositional learning have proceeded largely separately. In an effort to promote developments that bridge between the two fields, this article surveys their respective research landscapes and discusses existing and future connections between them.

## 1 Introduction

Consider the standard supervised machine learning (ML) setting. The learning agent receives a large labeled data set, and processes this entire data set with the goal of making predictions on data that was not seen during training. The central assumption for this paradigm is that the data used for training and the unseen future data are independent and identically distributed (*i.i.d.*). For example, a service robot that has learned a vision model for recognizing plates in a kitchen will continue to predict plates in the same kitchen. As AI systems become more ubiquitous, this *i.i.d.* assumption becomes impractical. The robot might move to a new kitchen with different plates, or it might need to recognize cutlery at a later time. If the underlying data distribution changes at any point in time, like in the robot example, then the model constructed by the learner becomes invalid, and traditional ML would require collecting a new large data set for the agent to learn to model the updated distribution. In contrast, a lifelong learning robot would leverage accumulated knowledge from having learned to detect plates and adapt it to the novel scenario with little data.

The lifelong learning problem is therefore that of learning under a nonstationary data distribution, making use of past knowledge when adapting to the updated distribution. One formalism for modeling the nonstationarity, which most existing works have adopted, is that of learning a sequence of distinct tasks—whether this formalism is adequate is not the focus of this survey and is left for a separate discussion. In the robot example, the tasks could be detecting plates in the first kitchen, detecting plates in the second kitchen, and detecting cutlery.

There are three main interdependent objectives of a lifelong learner:

- **Performing well on new tasks.** As discussed above, one requirement that a lifelong learner should satisfy is to accelerate the learning of future tasks by leveraging knowledge of past tasks. This ability, often denoted *forward transfer*, requires the agent to discover knowledge that is reusable in the future without knowing what that future looks like.

- **Performing well on recurrences of previous tasks.** A second requirement, which is typically in tension with the first, is that the agent should be capable of performing tasks seen in the past even long after having learned them. This requires the agent to maintain its previous knowledge, including retiring outdated knowledge as needed. For example, the robot might move back to the first kitchen after adapting to the second, and it should still be able to recognize plates. This objective necessitates that the agent avoids catastrophic forgetting (McCloskey and Cohen, 1989), but also that, whenever possible, it achieves *backward transfer* to those earlier tasks (Ruvolo and Eaton, 2013; Lopez-Paz and Ranzato, 2017). This is possible whenever knowledge from future tasks is useful for learning better models for older tasks. While most work views avoiding forgetting as a means solely for maintaining high performance on earlier tasks, in many cases it also permits better forward transfer, by enabling the agent to retain more general knowledge that works for a large set of tasks.

- **Supporting tractable long-term retention.** In addition to these two desiderata, the growth of the agent's memory use should be constrained over time. This choice is practical: if the agent requires storing large models or data sets for all past tasks in memory, then it would become impractical to handle very long task sequences. Note that, while it might be cheap to store huge amounts of data in memory, making use of such massive stored information would often incur an equivalently large computational cost, which is infeasible for many applications.

All these desiderata can be summarized as discovering knowledge that is *reusable*: reusable knowledge can be applied to both future and past tasks without uncontrolled growth. Beyond lifelong learning, the autonomous discovery of reusable knowledge has motivated work in transfer learning, multitask learning (MTL), and meta-learning—all of which deal with learning diverse tasks. These fields have received tremendous attention in the past decade, leading to a large body of literature spanning supervised learning, unsupervised learning, and reinforcement learning (RL). Traditionally, methods for solving this problem have failed to capture the intuition that, in order for knowledge to be maximally reusable, it must capture a self-contained unit that can be *composed* with similar pieces of knowledge. This compositionality refers to the ability of an agent to tackle parts of each problem individually, and then reuse the solution to each of the subproblems in combination with others to solve multiple bigger problems that contain shared parts. For example, a service robot that has learned to both search-and-retrieve objects and navigate across a university building should be able to quickly learn to put these together to deliver a stapler to Jorge's office. Instead, typical methods make assumptions about the way in which different tasks are related, and impose a structure that dictates the knowledge to share across tasks, usually in the form of abstract representations that are not explicitly compositional.

Yet, compositionality is a promising notion for achieving the three lifelong learning requirements listed above. Knowledge that is compositional can be used in future tasks by combining it in novel ways—this enables forward transfer. Further, not all knowledge must be updated upon learning new tasks to account for them, but only the components of knowledge that are used for solving these new tasks—this prevents catastrophic forgetting of any unused component and could enable backward transfer to tasks that reuse shared components. Finally, compositional knowledge permits solving combinatorially many tasks by combining that knowledge in different ways; conversely, solving a fixed number of tasks requires logarithmically many knowledge components, thereby inhibiting the agent's memory growth. This article focuses on a form of *functional composition*, exemplified in Figure 1, whereby each module or component processes an input and produces an output to be consumed by a subsequent module. The process is analogous to programming, where functions specialize to solve individual subproblems and combine to solve complex problems.

Recently, an increasing number of works have focused on the problem of learning compositional chunks of knowledge to share across different tasks (Andreas et al., 2016; Hu et al., 2017; Kirsch et al., 2018; Meyerson and Miikkulainen, 2018). At a high level, these methods aim to simultaneously discover *what* are the pieces

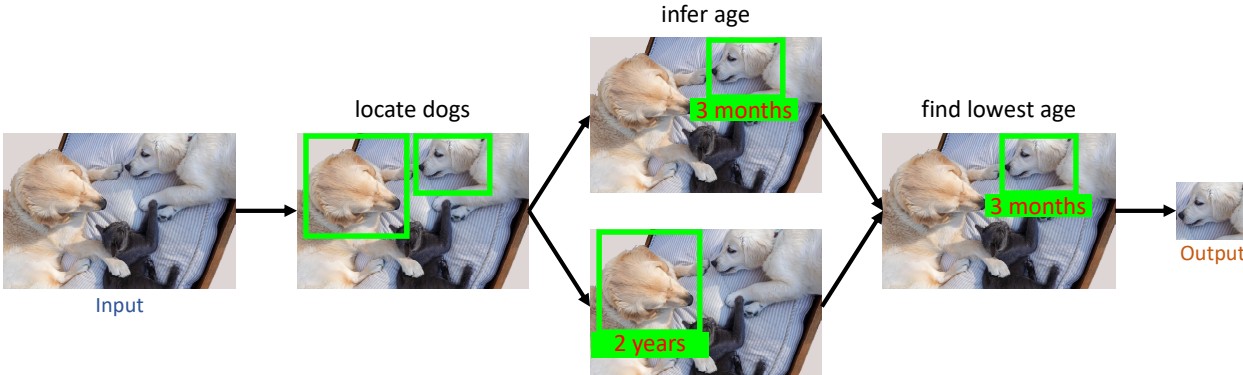

**Task:** locate youngest dog

Figure 1: Example of the functional composition studied in this article. The model decomposes the task "locate youngest dog" into simpler functions that are combined into an overall solution.

of knowledge to reuse across tasks and *how* to compose them for solving each individual task. Most studies in this field have made one of two possible assumptions. The first assumption is that the agent has access to a large batch of tasks to learn simultaneously in an MTL setting. This way, the agent can attempt numerous combinations of possible components and explore how useful they are for solving all tasks jointly. While this assumption simplifies the problem by removing the forward and backward transfer requirements, it is unfortunately unrealistic: AI systems in the real world will not have access to batches of simultaneous tasks, but instead will face them in sequence in a lifelong setting—agents may face several learning tasks simultaneously (e.g., vision, language, and decision-making modalities from a single data stream), leading to a form of mini-batch lifelong setting. The second assumption is that the agent does face tasks sequentially, but it is capable of learning components on a single task that are reusable for solving many future tasks. This latter assumption, albeit more realistic, relies on the ability to find optimal and reusable components from solving a single task, which is not generally possible given the limited data available for the task and the uncertainty about future components and how the knowledge must be compatible with them.

## 1.1 Article overview

This article reviews the existing literature with an aim to provide context for future works on the nascent field of lifelong compositionality. The survey is primarily a separate discussion of two topics that are closely related, yet previously disjointly studied: 1) lifelong or continual learning and 2) compositional knowledge representations. Research into lifelong learning seeks to endow agents with the capability to accumulate knowledge over a nonstationary stream of data, typically presented to the agent in the form of tasks. In principle, if the tasks are related in some way, the agent should be able to detect and extract the commonalities across the tasks in order to leverage shared knowledge and improve its overall performance. On the other hand, the goal of learning compositional knowledge representations is to decompose complex problems into simpler subproblems, such that the solutions to the easier subproblems can be combined to solve the original, harder problem. This formulation makes compositional representations an appealing mechanism for learning the relations across multiple tasks: by discovering subproblems that are common to many tasks; the learner could reuse the solutions to these subproblems as modules that compose in different combinations to solve the many tasks. Despite their intuitive appeal, few works have explicitly used compositional representations as a means for transferring knowledge across a lifelong sequence of tasks.

The discussion further divides works into those carried out in the supervised and the RL settings. While many of the techniques used for one are applicable to the other (albeit with minor-to-major adaptations), research into these two fields has proceeded mostly separately, with the vast majority of works focusing on the supervised setting. In particular, the form of functional composition studied in this article had been almost entirely overlooked in the RL literature until very recently.

| 1. Introduction | |
|---|---|
| 2. Problem formulation | |
| 3. Categorization of lifelong compositional approaches | |
| **Supervised/Unsupervised** | **Reinforcement Learning** |
| **Non-compositional** 4. Lifelong or continual learning | 6. Lifelong reinforcement learning |
| **Compositional** 5. Compositional knowledge | 7. Compositional reinforcement learning |
| 8. Summary and directions for future work | |

Figure 2: Organization of this article.

The discussion ties works together by categorizing them along six axes (detailed in Section 3). The first axis divides works according to the *learning setting*: lifelong learning, MTL, and single-task learning (STL). The second axis analyzes each approach according to whether the environment provides the *structure* of the task to the agent and how—this refers to the compositional structure in compositional works, and to the relations among tasks in more general lifelong settings. The third axis dissects works in terms of the underlying *learning paradigm* in which the agent is evaluated: supervised, unsupervised, or RL. The fourth axis separates approaches according to the *type of compositionality* they study: functional composition, temporal composition, or no composition. The fifth axis classifies works with respect to *how they structurally combine components*: via chaining, aggregation, or a more general graph. The sixth axis divides works in terms of the *application domain* they consider.

Multiple recent surveys have reviewed aspects of the continual learning problem, with special focus on the supervised setting (Parisi et al., 2019; De Lange et al., 2022). Closer to this article, Hadsell et al. (2020) included a section on modular architectures, and Khetarpal et al. (2020) reviewed existing works and open directions in lifelong RL. Other surveys have focused on the use of lifelong learning for specific applications, such as vision (Qu et al., 2021), language (Biesialska et al., 2020), and robotics (Lesort et al., 2019). In a similar spirit to this survey, Mundt et al. (2022) presented the CLEVA-Compass, a visual representation for analyzing lifelong learning approaches according to a variety of axes. Unlike the axes in this work, which focus on properties of the methods, the axes of the CLEVA-Compass measure properties of the evaluation protocols applied to those methods. This survey emphasizes peer-reviewed work from ML conferences, focusing on the years from 2017, with appropriate context from earlier literature and non-peer-reviewed work where appropriate. Compositionality has been examined in many other fields, including vision, language, and broader AI, with many overlapping ideas, but this review focuses on the (lifelong) ML perspective.

The remainder of this article is organized as follows, as depicted graphically in Figure 2. Section 2 formulates the problems of lifelong learning and compositional learning, providing examples of the types of tasks that each formulation accepts. Section 3 details the categorization of approaches used throughout the article and includes a visual depiction of the research landscape along the six axes. Section 4 surveys the literature on lifelong learning focused on the supervised case, dividing works into task-agnostic and task-aware, and expanding on techniques based on regularization, replay, generative replay, and capacity expansion. Section 5 discusses existing works that learn to functionally decompose supervised learning problems, placing particular emphasis on approaches that rely on modular neural architectures; this section also includes a summary of the few existing compositional works that have operated in the lifelong setting. Section 6 reviews methods for lifelong RL, which remains to date a substantially underdeveloped field. Section 7 discusses the large variety of forms of compositionality that have been proposed in RL, where notably only a handful of techniques functionally decompose learning problems. Finally, Section 8 summarizes the findings of this article and proposes high-impact avenues for future work to investigate.

## 2 Problem formulation

Despite their intuitive connections, lifelong learning and compositional learning have largely proceeded as disjoint lines of work. This section describes the concrete problem formulations for both lifelong learning

and compositional learning as studied in this article. At a high level, lifelong learning is the problem of accumulating knowledge over time and reusing it to solve related tasks, while compositional learning is the problem of decomposing knowledge into maximally reusable components.

## 2.1 The lifelong learning problem

At the highest level, lifelong learning involves learning over a nonstationary (non-*i.i.d.*) and potentially never-ending stream of data. From this high-level definition, different works have proposed multiple concrete instantiations of the problem. This section dissects common problem formulations in the literature, describes some of the advantages and disadvantages of each formulation, and provides example problems that can be captured under the most widely used definition.

Van de Ven and Tolias (2019) categorized lifelong learning problem definitions in terms of how nonstationarity is presented to the agent, proposing the following three variations:

- **Task-incremental learning.** This is the most common problem definition, which introduces nonstationarity into the learning problem in the form of *tasks*. Each task $\mathcal{Z}^{(t)}$ is itself a standard *i.i.d.* learning problem, with its own input space $\mathcal{X}^{(t)}$ and output space $\mathcal{Y}^{(t)}$. There exists a ground-truth mapping $f^{(t)} : \mathcal{X}^{(t)} \mapsto \mathcal{Y}^{(t)}$ that defines the individual task, as well as a cost function $\mathcal{L}^{(t)}\left(\hat{f}^{(t)}\right)$ that measures how well a learned $\hat{f}^{(t)}$ matches the true $f^{(t)}$ under the task's data distribution $\mathcal{D}^{(t)}\left(\mathcal{X}^{(t)}, \mathcal{Y}^{(t)}\right)$. During the learning process, the agent faces a sequence of tasks $\mathcal{Z}^{(1)}, \dots, \mathcal{Z}^{(t)}, \dots$. The learner receives a data set $\boldsymbol{X}^{(t)}, \boldsymbol{Y}^{(t)} \sim \mathcal{D}^{(t)}\left(\mathcal{X}^{(t)}, \mathcal{Y}^{(t)}\right)$ along with a task indicator $t$ that identifies the current task, but not how it relates to other tasks. Upon facing the $t$-th task, the goal of the learner is to solve (an online approximation of) the MTL objective:

$$z_t = \frac{1}{t} \sum_{\hat{t}=1}^{t} \mathcal{L}^{(\hat{t})}\left(\hat{f}_t^{(\hat{t})}\right) \; , \tag{1}$$

  where $\hat{f}_t^{(\hat{t})}$ is the predictor for task $\mathcal{Z}^{(\hat{t})}$ at time $t$. Note that this definition assumes that the predictors for earlier tasks are affected by the learning of subsequent tasks (as indicated by the subscript $t$), but makes no explicit assumptions about how the predictors are related (e.g., a single shared model or a shared representation with task-specific predictors on top of the representation).

- **Domain-incremental learning.** The key distinguishing factor of this setting is that the learning problem does not inform the agent of the task indicator $t$. Instead, the tasks vary only in their input distribution $\mathcal{D}^{(t)}\left(\mathcal{X}^{(t)}\right)$, but there exists a single common solution that solves all tasks. For example, the problem could be a binary classification problem between "cat" and "dog", and each different task could be a variation in the input domain (e.g., changing light conditions or camera resolutions). The goal of the learner is still to optimize the approximate MTL objective of Equation 1.

- **Class-incremental learning.** In this setting, there is a single multiclass classification task with a large number of classes (i.e., $\mathcal{Y} = \{1, \dots, C\}$ for some large $C$). The agent observes classes sequentially, and must be able to predict the correct class among all previously seen classes. For example, the task could be ImageNet (Deng et al., 2009) classification, and classes could be presented to the agent ten at a time, with later stages not containing previous classes in the *training* data, but indeed requiring accurate prediction across all seen classes in the *test* data. One alternative way to define this same problem is that each learning stage (i.e., each group of classes) is a distinct task, and the goal of the agent is to simultaneously predict the current task indicator $t$ and the current class within that task $\mathcal{Z}^{(t)}$. The latter equivalent formulation, though less intuitive, enables framing the learning objective in exactly the same way as the previous two problem settings, per Equation 1.

Most existing works have considered the task-incremental setting. Concretely, a vector $\boldsymbol{\theta}^{(t)}$ parameterizes each task's solution, such that $\hat{f}^{(t)} = f_{\boldsymbol{\theta}^{(t)}}$. After training on $T$ tasks, the goal of the lifelong learner is to

find parameters $\boldsymbol{\theta}^{(1)}, \ldots, \boldsymbol{\theta}^{(T)}$ that minimize the cost across all tasks: $\frac{1}{T} \sum_{t=1}^{T} \mathcal{L}^{(t)}\left(\hat{f}^{(t)}\right)$. The agent does not know the total number of tasks, the order in which tasks will arrive, or how tasks are related to each other.

Given a limited amount of data for each new task, typically insufficient for obtaining optimal performance in isolation, the agent must strive to discover any relevant information to 1) relate it to previously stored knowledge in order to permit transfer and 2) store any new knowledge for future reuse. The environment may require the agent to perform any previous task, implying that it must perform well on *all* known tasks at any time. In consequence, the agent must strive to retain knowledge from even the earliest learned tasks.

While prior work in supervised learning has described this task-incremental setting as artificial, it contains some desirable properties that are missing from other common definitions. First, unlike in domain-incremental learning, it is not necessary that the inputs are the only aspect that changes over time. This is useful when extending these definitions to the RL setting, where different tasks naturally correspond to different reward functions or transition dynamics. Second, unlike in class-incremental learning, extension to RL is straightforward, since the learning objective can still easily be averaged across clearly differentiated tasks.

All the above definitions assume that the agent is required to perform well on all previously seen tasks, which is often not realistic. For example, consider a service robot that has provided assistance for a long time in a small one-floor apartment, and is later moved to a much larger two-floor home. While knowledge from the small apartment may be useful for quickly adapting to the larger home, over time retaining full knowledge about how to traverse the small apartment might become counterproductive as that information becomes obsolete. Therefore, a handful of works have developed techniques for nonstationary lifelong learning, where not only the *data* distribution changes from task to task, but also the distribution over *tasks* itself changes from time to time (like in the small-apartment-to-large-home example). In this setting, the objective must change, since it is no longer desirable to retain performance on tasks from outdated distributions. However, works in this line fall outside of the scope of this survey and are left for a separate discussion.

### 2.1.1 Evaluation settings and performance metrics

Due to this survey's focus on the intersection of lifelong learning and functional composition, we only briefly touch on evaluation metrics and methodologies specific to lifelong learning. For a more detailed discussion, see Section 3.5 in the work of Parisi et al. (2019).

In the task-incremental supervised setting, the standard way to craft the different tasks for benchmarking purposes originates from the class-incremental setting: split each data set into multiple smaller tasks, each containing a subset of the classes. As an example, CIFAR-100 (Krizhevsky and Hinton, 2009), a visual recognition data set of 100 classes corresponding to various object types, is commonly split randomly into 20 individual 5-way classification tasks to evaluate lifelong agents. However, to show that this is not the only possible setting that the proposed methods can handle, some evaluations have also executed experiments in a more complex setting with tasks from various distinct data sets. Most recently, Bornschein et al. (2022) released such a multi-data-set benchmark with 100 tasks sampled from 30 years of computer vision research. Typical measures used to evaluate the performance of lifelong learning methods include, among others, overall accuracy, forward transfer, backward transfer, and jumpstart performance.

Note that, while the benchmarks used in the lifelong learning literature likely contain some level of implicit composition, they do not permit explicit analysis of compositionality. In consequence, researchers interested in studying the joint problem posed in this article should likely consider carrying out evaluations both on standard lifelong learning benchmarks, for adequate placement in the literature, and on explicitly compositional benchmarks, for rigorous analysis of the compositionality of their approaches.

In the case of RL, most approaches are evaluated on custom benchmarks. Section 7.1 summarizes existing works seeking to standardize evaluation practices for multitask and lifelong RL, with a focus on compositionality.

### 2.2 The compositional learning problem

Section 2.1 described the lifelong learning problem in terms of how nonstationarity is presented to the agent in the form of a sequence of tasks. However, it does not provide insight into how the different tasks might be

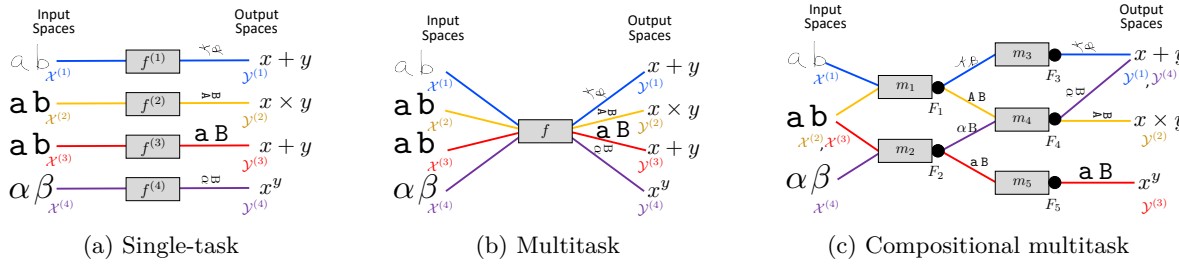

(a) Single-task            (b) Multitask            (c) Compositional multitask

Figure 3: Compositional problem graphs. Each node in the graph represents a random variable for a representational space, produced by the output of a module or function. STL agents assume that tasks are unrelated and learn modules in isolation, while monolithic MTL agents assume that all tasks share a single module. In contrast, more general compositional MTL agents assume that tasks selectively share a set of modules, yielding different solutions to each task constructed from common solutions to subtasks.

related to each other. In particular, this article focuses on tasks that are *compositionally* related and methods that explicitly exploit these compositional assumptions.

Following the problem formulation from Chang et al. (2019), compositional methods assume (either implicitly or explicitly) that each task can be decomposed into subtasks. Equivalently, the predictive function $f^{(t)}$ characterizing each task can be decomposed into multiple subfunctions $F_1^{(t)}, F_2^{(t)}, \ldots$, such that $f^{(t)} = F_1^{(t)} \circ F_2^{(t)} \circ \cdots \left( \boldsymbol{x}^{(t)} \right)$. This assumption trivially holds for any function $f^{(t)}$. Critically, the formulation further assumes that there exists a set of $k$ subfunctions that are common to all tasks the agent might encounter: $F_i^{(t)} \in \{F_1, \ldots, F_k\} \ \forall t, i$, such that there is potential for compositional reuse across tasks.

This way, the full learning problem can be characterized by a directed graph $\mathcal{G} = (\mathcal{V}, \mathcal{E})$, with two types of nodes. The first type represents the inputs and outputs of each task as random variables. Concretely, each task has an input node $u^{(t)}$ with in-degree zero and an output node $v^{(t)}$ with out-degree zero such that $u^{(t)}, v^{(t)} \sim \mathcal{D}^{(t)} \left( \mathcal{X}^{(t)}, \mathcal{Y}^{(t)} \right)$. The second type of nodes $F$ represents functional transformations such that:

1. for every edge $\langle u, F \rangle$ the function $F$ takes as input the random variable $u$,

2. for every edge $\langle F, F' \rangle$ the output of $F$ feeds into $F'$, and

3. for every edge $\langle F, v \rangle$ $v$ is the output of $F$.

With this definition, the paths in the graph from $u^{(t)}$ to $v^{(t)}$ represent all possible solutions to task $\mathcal{Z}^{(t)}$ given a set of functional nodes.

This formalism also has an equivalent generative formulation. In particular, a compositional function graph $\mathcal{G}$ generates a task $\mathcal{Z}^{(t)}$ by choosing one input node $u^{(t)}$ and a path $p^{(t)}$ through the graph to some node $v^{(t)}$. Then, the following two steps define the generative distribution for task $\mathcal{Z}^{(t)}$. First, instantiate the random variable $u^{(t)}$ by sampling from the input distribution $u^{(t)} = \boldsymbol{x}^{(t)} \sim \mathcal{D}^{(t)}$. Next, generate the corresponding label $\boldsymbol{y}^{(t)}$ by compositionally applying all functions in the chosen path $p^{(t)}$ to the sampled $u^{(t)}$. As noted by Chang et al. (2019), there are generally multiple possible compositional solutions to each task. One additional reasonable assumption is that the generative problem graph is that with the minimum number of possible nodes, such that nodes (i.e., subtasks) are maximally shared across different tasks. This choice intuitively implies the maximum amount of possible knowledge transfer across tasks.

Figure 3 shows three different assumptions that learning algorithms make over the space of tasks. The left-most graph (Figure 3a) shows the standard STL assumption: each task $\mathcal{Z}^{(t)}$ is completely independent from the others, and therefore the agent learns the predictive functions $\hat{f}^{(t)}$ in isolation. Note that this doesn't explicitly prohibit learning compositional solutions: each $\hat{f}^{(t)}$ could itself be decomposed into multiple subtasks, but the subtasks would still be individual to each task. The center graph (Figure 3b) shows the typical monolithic MTL assumption: all different tasks can be solved with a single common solution. The

right-most graph (Figure 3c) shows the assumption made by compositional approaches: each task can be decomposed into a task-specific sequence of subtasks, but the set of possible subtasks is common to all tasks.

As a first example that matches the latter formulation, consider the following set of tasks:

- $\mathscr{Z}^{(1)}$: count the number of cats in an image

- $\mathscr{Z}^{(2)}$: locate the largest cat in an image

- $\mathscr{Z}^{(3)}$: locate the largest dog in an image

- $\mathscr{Z}^{(4)}$: count the number of dogs in an image

These tasks can be decomposed into: detect cats, detect dogs, locate largest, and count. If an agent learns tasks $\mathscr{Z}^{(1)}$, $\mathscr{Z}^{(2)}$, and $\mathscr{Z}^{(3)}$, and along the way discovers generalizable solutions to each of the four subtasks, then solving $\mathscr{Z}^{(4)}$ would simply involve reusing the solutions to the dog detector and the general counter.

Consider another example from the language domain, with tasks given by:

- $\mathscr{Z}^{(1)}$: translate text from English to Spanish

- $\mathscr{Z}^{(2)}$: translate text from Spanish to Italian

- $\mathscr{Z}^{(3)}$: translate text from English to Italian

As above, if the learner has learned to solve tasks $\mathscr{Z}^{(1)}$ and $\mathscr{Z}^{(2)}$, it could solve task $\mathscr{Z}^{(3)}$ by first translating the text from English to Spanish and subsequently translating the resulting text from Spanish to Italian. Here, Spanish would act as a *pivot language* (Boitet, 1988).

This definition can be applied to RL problems as well. Consider the following task components in a robotic manipulation setting:

- Robot manipulator: diverse robotic arms with different dynamics and kinematic configurations can be used to solve each task.

- Objective: each task might have a different objective, like placing an object on a shelf or throwing it in the trash.

- Obstacle: various obstacles may impede the robot's actions, such as a door frame the robot needs to go through or a wall the robot needs to circumvent.

- Object: different objects require different grasping strategies.

One way to solve each robot-objective-obstacle-object task is to decompose it into subtasks: detect the grasping points of the object, detect regions unobstructed by the obstacle, plan a trajectory for reaching the objective, and drive the robot's joints. Note that this is not equivalent to the temporal composition of skills or options. Instead, each time step requires solving all subtasks simultaneously (e.g., the actions must be tailored to the current robot arm at all times). Mendez et al. (2022b) provide a precise description of the problem formulation for the RL case, where the (more general) subproblems may involve sensing, acting, or a combination of both. Section 7 further discusses the two formulations.

There is no consensus in the research community for a set of evaluation procedures or measures to assess the quality of the compositional solutions discovered by existing methods, in part due to the lack of precise definitions of compositionality. Sections 5 and 7 discuss the ways existing works have evaluated approaches in the supervised and the RL settings, respectively, as well as methods developed specifically to understand the compositional qualities of these approaches' solutions.

# 3   Categorization of lifelong compositional approaches

This section describes in detail the six axes used to categorize existing methods. By design, this categorization bridges between lifelong learning approaches and compositional approaches, laying the foundations for the field to explore new techniques at the intersection of the two.

1. **Multiplicity of tasks.** *Single-task* methods train models individually on a single task, without any notion of shared knowledge or transfer. While this is the focus of most ML research, it is not the focus of this survey. However, many compositional methods, particularly in RL, seek to simultaneously extract the compositional structure and learn a policy within a single task. The focus of this survey is on methods that share information across multiple tasks. In particular, *multitask* approaches train on multiple tasks simultaneously, while *lifelong* methods train on tasks one after another in sequence. Very few approaches have trained compositional models in a lifelong setting. Note that this survey categorizes approaches as lifelong methods if they face tasks in sequence regardless of the underlying mechanism they use for training (e.g., maintaining separate models for each task with some shared parameters). In addition, the categorization of single-task and multitask approaches is not always clear, since some methods might treat solving multiple tasks as a single-task generalization problem; therefore, we mainly follow the language used in each paper to guide this classification.

2. **Knowledge about task structure.** Methods that assume that the *structure is explicitly given* to the agent do so in the form of external inputs. This might take the form of task descriptors, pretrained modules, or hard-coded modular structures. Approaches that consider *implicitly given structure* expect the input itself (e.g., image or language prompt) to contain cues that either distinguish tasks from each other (in noncompositional works) or that inform appropriate compositional solutions to each task (in compositional works). In most cases, techniques in these two categories do not assume that a task indicator is provided, since the input itself contains sufficient information to extract it. On the other hand, methods that assume *no access to the task structure* rely on some form of search to identify how tasks relate to one another. Most commonly, these approaches do so by relying on task indicators that enable them to learn different models for each task.

3. **Learning paradigm.** This axis considers whether the problems faced by the agent are *supervised*, *unsupervised*, or *RL* problems. Concretely, this is not fundamentally tied to the training mechanism used by the agent, but to the underlying problem. For example, multiple algorithms for learning compositional architectures use RL techniques to solve the non-differentiable search, but themselves solve supervised learning problems; such approaches are categorized as supervised.

4. **Type of composition.** Most of ML, and in particular most of lifelong learning, considers *noncompositional* problems. This survey considers a method to be compositional if it uses a compositional model (e.g., modular neural networks) or if it seeks to solve a compositional problem (e.g., compositional zero-shot generalization). While all supervised learning methods consider *functional* composition, RL methods can consider functional or *temporal* composition. This distinction, which refers to whether components are chained in time or in function space, is described in detail in Section 7.

5. **Type of structural configuration.** For any compositional method, we consider three types of structural configurations. *Aggregation* combines solutions in a flat manner, without any notion of hierarchy. For example, methods for logical composition in RL often aggregate models by summing their value functions. *Chaining* combines components hierarchically, with one component being executed after the next, and with a partial ordering over the components such that, if $a \prec b$, component $a$ must always come before component $b$ in the chained model. Arbitrary *graph* composition chains components hierarchically, too, but assumes no fixed ordering over them.

6. **Application domain.** This axis specifically considers the domains to which a paper applies its proposed method. Methods in this survey have been evaluated in *vision*, *robotics*, *language*, *visual question answering* (VQA), *audio*, and *toy* domains.

Figure 4 includes a visual depiction of the landscape according to this categorization, while Appendix A lists all cited references in terms of the same categories.

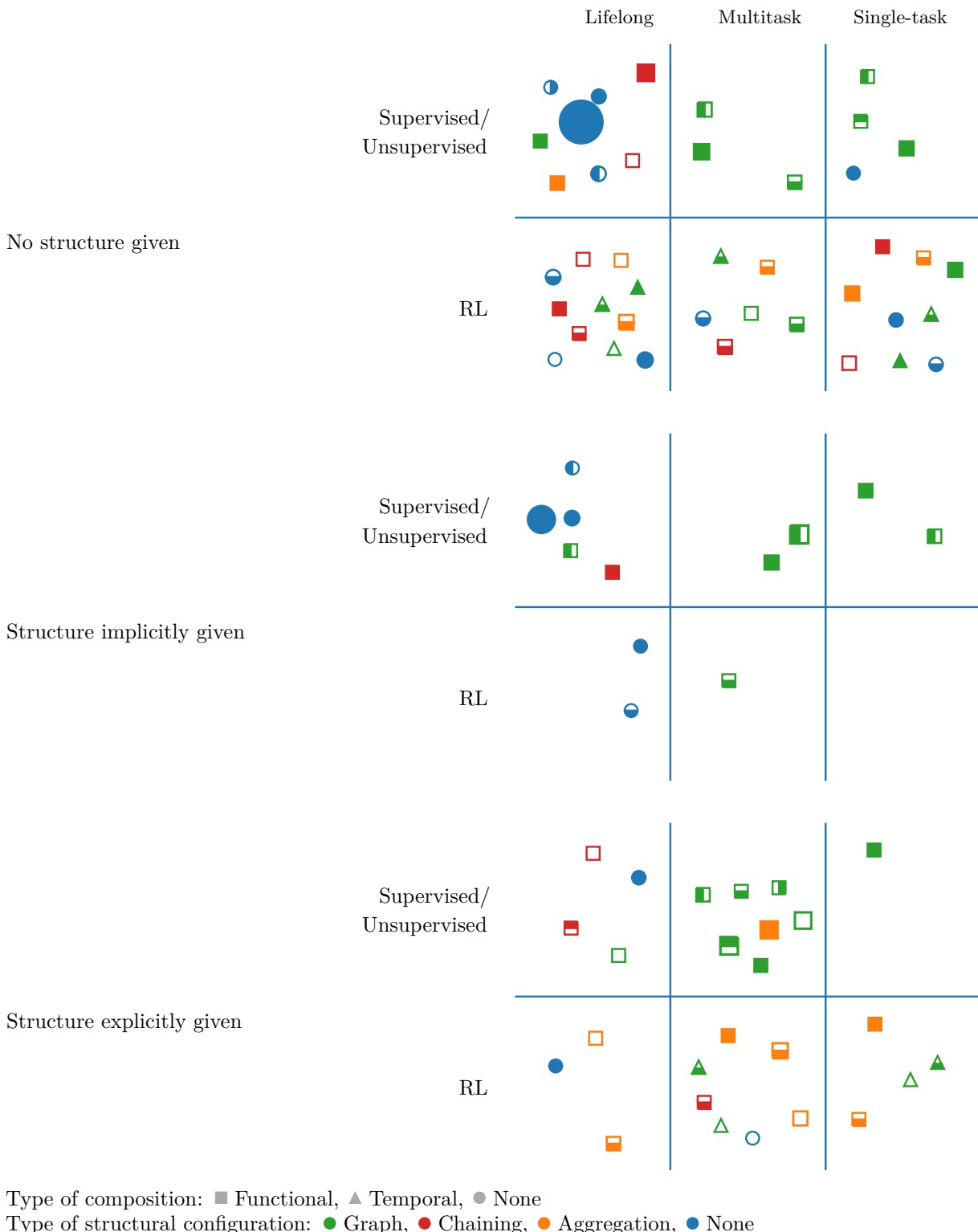

Type of composition: ■ Functional, ▲ Temporal, ● None
Type of structural configuration: ● Graph, ● Chaining, ● Aggregation, ● None
Application domain: ● Vision, ◗ Robotics, ◑ Language, ◒ VQA, ◐ Audio, ○ Toy

Figure 4: A categorization of existing works into six axes, defined in text within the figure. The scale of the icon represents the number of references in this survey per category. Most work on lifelong or continual learning has not learned explicitly compositional structures, while most efforts on compositional learning have operated in the MTL or STL settings. Appendix A contains a tabular version of this figure, listing all references in their respective categories. Best viewed in color.

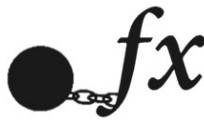

### Regularization
Avoid interference with previous tasks

*Section 4.1.1*
- *functional regularization*
- *preserve important parameters*
- *orthogonal model subspaces*

*Section 4.2.1*
- *Bayesian models*
- *trust regions of previous tasks*

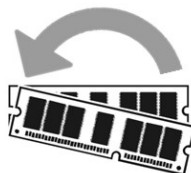

### Replay
Recall stored data representative of past tasks

*Section 4.1.2*
- *experience replay*
- *gradient episodic memory*

*Section 4.2.2*
- *memory-based task similarity*
- *selective storage of memories*

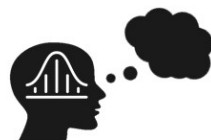

### Generative Replay
Hallucinate data for replay

*Section 4.1.3*
- *generative adversarial nets*
- *generative task models*

*Section 4.2.3*
- *dynamic mixture models*
- *parameter and data generators*

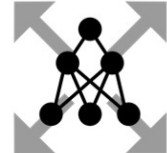

### Expandable Models
Expand model capacity to handle new tasks

*Section 4.1.4*
- *dynamically expandable nets*
- *generative task models*

*Section 4.2.4*
- *dynamic mixture of experts*
- *hierarchical Bayesian models*

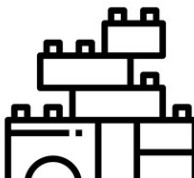

### Reusable Knowledge
Learn and transfer reusable chunks of knowledge

*Section 4.3*
- *factorized transfer / online dictionary learning*
- *selective layer reuse in deep nets*
- *selective transfer via attention*

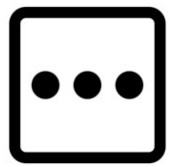

### Other Approaches

*Section 4.4*
- *online meta-learning*
- *continual generalized zero-shot learning*
- *stable stochastic gradient descent*
- *recurrent neural nets*

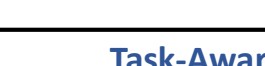

**Task-Aware**          **Task-Agnostic**

Figure 5: Major mechanisms used for lifelong or continual learning and techniques exemplifying those mechanisms in both task-aware and task-agnostic settings. Best viewed in color.

# 4 Lifelong or continual learning

Lifelong learning agents face a variety of tasks over their lifetimes, and should accumulate knowledge in a way that enables them to more efficiently learn to solve new problems. Thrun (1998) first introduced the concept of lifelong learning, which has received widespread attention in recent years (Chen and Liu, 2018). Recent efforts have mainly focused on avoiding catastrophic forgetting (McCloskey and Cohen, 1989). At a high level, existing approaches define parts of parametric models (e.g., deep neural networks) to share across tasks. As the agent encounters tasks sequentially, it strives to retain the knowledge required to solve earlier tasks.

The following sections divide lifelong learning methods into task-aware (with task indicator) and task-agnostic (without task indicator), examining each according to the major mechanisms used. A summary of approaches according to this division is shown in Figure 5. The discussion then ties the task-awareness division back to the separation according to how models receive information about the structure of the tasks.

## 4.1 Task-aware

The majority of works in lifelong learning in recent years fall in the category of task-aware methods. This section summarizes the existing literature in this category. Note that most methods that can operate in the task-aware setting can in principle also operate in the task-agnostic setting. In such cases, their categorization into task-aware or task-agnostic follows the (majority of the) experiments used in their original evaluation.

### 4.1.1 Regularization

One common approach to avoid forgetting is to impose data-driven regularization to prevent parameters from deviating in directions that are harmful to performance on the early tasks. The intuition is that similar parameters would lead to similar solutions to the earlier tasks. Formally, these approaches approximate the MTL objective in Equation 1 with a data-driven regularization term:

$$z_t(\boldsymbol{\theta}) = \frac{1}{t}\mathcal{L}\Big(\boldsymbol{X}^{(t)}, \boldsymbol{Y}^{(t)}, \boldsymbol{\theta}\Big) + \frac{1}{t}\sum_{\hat{t}=1}^{t-1}\Omega\Big(w_{\hat{t}}, \boldsymbol{\theta}, \boldsymbol{\theta}_{\hat{t}}^{(\hat{t})}\Big) \ , \tag{2}$$

where $w_{\hat{t}}$ are regularization weights obtained from the data of task $\mathcal{Z}^{(\hat{t})}$, $\boldsymbol{\theta}$ are the parameters being optimized, and $\boldsymbol{\theta}_{\hat{t}}^{(\hat{t})}$ are the parameters obtained at time $\hat{t}$. The canonical example of this idea is *elastic weight consolidation* (EWC), which, inspired by a Bayesian formulation, places a quadratic penalty on the parameters for deviating from the parameters of each previously seen task, weighted by the diagonal of the Fisher information matrix of each task, $\boldsymbol{F}^{(t)}$: $\Omega(\boldsymbol{\theta}) = \sum_{\hat{t}}\big(\boldsymbol{\theta} - \boldsymbol{\theta}^{(\hat{t})}\big)^{\top}\boldsymbol{F}^{(\hat{t})}\big(\boldsymbol{\theta} - \boldsymbol{\theta}^{(\hat{t})}\big)$ (Kirkpatrick et al., 2017). EWC is one of the few exceptional works that was applied to both supervised and RL settings. An extension of EWC uses a Kronecker-factored approximation of the Fisher information matrix instead of a diagonal to improve performance at little additional cost (Ritter et al., 2018).

Following this same principle, the literature has proposed a variety of regularization terms. Departing from the Bayesian formulation, the *synaptic intelligence* approach of Zenke et al. (2017) computes an estimate of each parameter's importance based on the trajectory of updates to the parameter, and uses this as the weighting for quadratic regularization. A generalized regularizer combines this latter idea with EWC (Chaudhry et al., 2018). Another well-known mechanism, *hard attention to the task*, learns task-specific, (nearly-)binary attention masks to select which nodes in a neural network to use for each task, and then constrains updates to parameters based on the previous tasks' masks (Serrà et al., 2018). A similar technique additively decomposes each task's parameters into a shared set of parameters modulated by a task-specific mask and a set of task-adaptive parameters, applying quadratic regularization to prevent previous tasks' parameters from diverging from their original values (Yoon et al., 2020). Recent work proposed a sparsity-based regularizer that reduces the storage space for regularization terms by computing node-wise (as opposed to parameter-wise) importance weights (Jung et al., 2020), and applied this approach to supervised and RL. Cha et al. (2021) proposed a complementary regularizer based on entropy maximization, which encourages wider local minima and can therefore be used in combination with existing regularizers to avoid forgetting.

The approaches above, originally applied to vision domains, have been extended to image captioning, demonstrating their applicability to language domains (Del Chiaro et al., 2020).

The methods described so far are based on the intuition that parameters that are important for previous tasks should be modified sparingly to avoid forgetting. Recent works have considered a different intuition: in order for new tasks to be learned without interfering with past tasks, they should lie on orthogonal subspaces of the parameter space. One mechanism for imposing orthogonality is to use precomputed task-specific orthogonal matrices to project the feature space of each task (Chaudhry et al., 2020). Alternatively, it is also possible to compute such projection matrices sequentially based on the learned solutions to previous tasks. This can be achieved by exploiting the singular vector space of the activations of the network, which applies to linear and convolutional layers (Saha et al., 2021; Deng et al., 2021), as well as recurrent layers (Duncker et al., 2020).

Another regularization strategy that has become popular is online variational inference, which approximates the Kullback-Leibler (KL) divergence between the current and previous predictive distributions in a Bayesian setting. Nguyen et al. (2018) developed the first *variational continual learning* (VCL) method, which, akin to EWC (Kirkpatrick et al., 2017) in the standard regularization-based setting, requires storing penalty terms for each network parameter. In an effort to reduce storage requirements, Ahn et al. (2019) modified this method by storing penalty terms for each network node, akin to the work of Jung et al. (2020). A generalized variational objective adds a tunable hyperparameter to weight the KL divergence term in the objective function, encompassing EWC and VCL (Loo et al., 2021). These notions can extend to Gaussian mixture distributions. In particular, Zhang et al. (2021) developed a continual variational inference method using a Chinese restaurant process to automatically determine the number of latent components, while Kumar et al. (2021) did the same using an Indian buffet process for unsupervised and supervised learning.

Other methods instead functionally regularize the outputs of the model (e.g., *learning without forgetting*, Li and Hoiem, 2017), penalizing deviations from the predictions of earlier tasks' models by optimizing for:

$$z_t(\boldsymbol{\theta}) = \frac{1}{t}\mathcal{L}^{(t)}\Big(\boldsymbol{X}^{(t)}, \boldsymbol{Y}^{(t)}, \boldsymbol{\theta}\Big) + \frac{1}{t}\sum_{\hat{t}=1}^{t-1}\Omega\Big(w_{\hat{t}}, \hat{f}_t^{(\hat{t})}, \hat{f}_{\hat{t}}^{(\hat{t})}\Big) \ , \tag{3}$$

where $\hat{f}_t^{(\hat{t})}$ and $\hat{f}_{\hat{t}}^{(\hat{t})}$ are the predictive functions for task $\mathcal{Z}^{(\hat{t})}$ obtained at times $t$ and $\hat{t}$, respectively. Critically, functional regularization requires evaluating the $\hat{f}$'s, which previous works have accomplished either via replay (as described in Section 4.1.2) or by applying those functions to the current task's data, $\boldsymbol{X}^{(t)}$. Benjamin et al. (2019) noted that distance in parameter space (parameter regularization) is not representative of distance in function space (functional regularization), which should be minimized to avoid forgetting. The authors then showed that naïvely estimating function-space distance on a small set of stored samples results in a strong functional regularization approach. Other methods convert the learned network into a Gaussian process (GP) and store a subset of samples from previous tasks in memory (Titsias et al., 2020; Pan et al., 2020). Then, the GP posterior on those points penalizes the model for making incorrect predictions on past tasks.

A recent approach uses EWC with the Fisher information of the current task only, to forget knowledge of past tasks that prevents further learning (Wang et al., 2021); this method was used in supervised and RL tasks.

Techniques in this category primarily focus on avoiding negative backward transfer by retaining models that are similar to the original models, while forward transfer might be enabled implicitly via parameter sharing.

### 4.1.2 Replay

A distinct approach retains a small buffer of data from all tasks, and continually updates the model parameters using data from the current and previous tasks, thereby maintaining the knowledge required to solve the earlier tasks. The general form of the objective for replay-based techniques can be written as:

$$z_t(\boldsymbol{\theta}) = \frac{1}{t} + \mathcal{L}(\boldsymbol{X}^{(t)}, \boldsymbol{Y}^{(t)}, \boldsymbol{\theta}) + \frac{1}{t}\sum_{\hat{t}=1}^{t-1}\Omega\Big(\boldsymbol{X}_{\text{rep}}^{(\hat{t})}, \boldsymbol{Y}_{\text{rep}}^{(\hat{t})}, \boldsymbol{\theta}\Big) \ , \tag{4}$$

where $\boldsymbol{X}_{\text{rep}}^{(\hat{t})}, \boldsymbol{Y}_{\text{rep}}^{(\hat{t})} = \texttt{buffer}\Big(\boldsymbol{X}^{(\hat{t})}, \boldsymbol{Y}^{(\hat{t})}\Big)$ are replay buffers given by some mechanism $\texttt{buffer}$ (typically a subsampling mechanism). Most naïvely, one could simply iterate over past tasks' data when training on the

new task—this is known as *experience replay* (ER, Chaudhry et al., 2019b), and corresponds to $\Omega = \mathcal{L}$. If the amount of data stored for replay is small, one would expect that the model could overfit to the tiny memory. However, empirical evaluations found that even this naïve replay approach performs surprisingly well.

Other popular approaches use replay data to constrain the directions of gradient updates to regions of the parameter space that do not conflict with the earlier tasks' gradients via *gradient episodic memory* (Lopez-Paz and Ranzato, 2017; Chaudhry et al., 2019a). These same techniques have been used with a meta-learning objective function, whereby the agent trains directly to optimize the network's feature representation to avoid gradient conflicts between future and previous tasks (Riemer et al., 2019; Gupta et al., 2020a).

Mirzadeh et al. (2021) developed a distinct objective function. The authors found that a linear curve in the objective function connects the solutions of the MTL and continual learning problems, and consequently used replay data to encourage finding a solution that is closer to the (no-forgetting) MTL solution. Similarly, Raghavan and Balaprakash (2021) found theoretically that the balance between generalization and forgetting, viewed as a two-player zero-sum game, is stable and corresponds to a saddle point, and developed an algorithm that searches for this saddle point by playing the two-player game.

Other works have not modified the objective function $\Omega$, but instead have focused on other aspects of the problem. For example, one approach balances the loss terms for replay and current data via mixed stochastic gradients (Guo et al., 2020b). As another example, Pham et al. (2021b) used a dual memory to train a set of shared neural network layers and a task-specific controller to transform the features of the shared layers.

Despite the appeal of these more advanced replay-based methods, the basic ER method that simply replays randomly stored data remains a strong and popular baseline (Chaudhry et al., 2019b). In practice, replay-based techniques have proven to be much stronger at avoiding forgetting than regularization-based methods. One potential theoretical explanation for this discrepancy is that optimally solving the continual learning problem requires storing and reusing all past data (Knoblauch et al., 2020).

Replay-based approaches have also followed other, less common directions. One nonparametric kernel method uses an episodic memory for detecting the task at inference time instead of for replay (Derakhshani et al., 2021). Other works have learned hypernetworks that take a task descriptor as input and output the parameters for a task-specific network, using replay at the task-descriptor level (von Oswald et al., 2020; Henning et al., 2021). In the unsupervised setting, Rostami (2021) used replay to learn a Gaussian mixture model in a latent representation space such that all tasks map to the mixture distribution in the embedding space.

Like regularization methods, approaches based on replay might achieve forward transfer as a result of parameter sharing. Intuitively, positive backward transfer could occur by virtue of repeated training over increasingly larger MTL problems, though most experimental results in the supervised setting have not exhibited this property. Instead, results have primarily focused on avoiding negative backward transfer.

### 4.1.3 Generative replay

A related technique is to replace the `buffer` mechanism by a generative model to "hallucinate" replay data, potentially reducing the memory footprint by avoiding storing earlier tasks' data. For example, this can be achieved by training a generative adversarial network (GAN) and using the trained network to generate artificial data for the previous tasks to avoid forgetting (Shin et al., 2017). In one of the few works that has considered lifelong language learning, the authors leveraged the intuition that a language model itself is a generative model and used it for replaying its own data (Sun et al., 2020). A recent unsupervised method for training GANs learns both global features that are kept fixed after the first task and task-specific transformations to those shared features (Varshney et al., 2021). While the approach does not train the GANs via generative replay, the learned GANs can generate replay data to train a supervised model.

### 4.1.4 Expandable models

While approaches described so far are capable of learning sequences of tasks without forgetting, they are limited by one fundamental constraint: the learning of many tasks eventually exhausts the capacity of the model, and it becomes impossible to learn new tasks without forgetting past tasks. Note that, while some replay and regularization approaches described so far add task-specific features that can be considered as a

form of capacity expansion, this expansion is naïvely executed for every task. Some additional methods use per-task growth as their primary mechanism for avoiding forgetting. One example specific to convolutional layers keeps the filter parameters fixed after initial training on a single task and adapts them to each new task via spatial and channel-wise calibration (Singh et al., 2020). However, these methods still consider the set of *shared* features to be nonexpansive, and these shared features might still run out of capacity. As one potential exception, another technique leverages large pretrained language models that in practice seem to have sufficient capacity for a massive number of tasks (specifically BERT, Devlin et al., 2019) and adds small modules trained via task-specific masking to learn a sequence of language tasks (Ke et al., 2021). A more recent mechanism that follows this same direction learns a continuous input vector per task to adapt the behavior of a large pretrained language model (Razdaibiedina et al., 2023).

To address capacity limitations, a similar line of work has studied how to automatically expand the capacity of the model as needed to accommodate new tasks. Yoon et al. (2018) trained *dynamically expandable networks* via a multistage process that first selects the relevant parameters from past tasks to optimize, then checks the loss on the new task after training, and—if the loss exceeds a threshold—expands the capacity and trains the expanded network with group sparsity regularization to avoid excessive growth. In order to avoid forgetting, the algorithm measures the change in each neuron's input weights, and duplicates the neuron and retrains it if the change is too large. A similar approach sidesteps the need for this duplication step by maintaining all parameters for past tasks fixed (Hung et al., 2019). A distinct method splits the capacity of the model across tasks selected via boosting, and adds one such model for every new task (Ramesh and Chaudhari, 2022).

Note that the focus of model capacity expansion is to avoid forgetting by permitting the model to use new weights to adapt to each new task. However, since new parameters are typically reserved for future tasks, previous tasks do not benefit from the additional capacity, preventing positive backward transfer.

### 4.1.5 Closing remarks on task-aware lifelong learning

Figure 4 categorizes the vast majority of the approaches discussed in this section as lifelong supervised learning methods with no composition, no task structure provided, and a focus on vision applications. A handful of exceptions were highlighted that deal with RL, unsupervised learning, or language applications.

While these techniques notably require no information of how tasks are related, they do require a task indicator during learning and evaluation. This choice enables task-aware methods to use task-specific parameters to specialize shared knowledge to each task, but may be inapplicable in some settings where there are no evident task boundaries or there is no potential for supervision at the level of the task indicator.

### 4.2 Task-agnostic

As an alternative, task-agnostic lifelong approaches automatically detect tasks. While the learning techniques are typically not fundamentally different from those for the task-aware setting, this survey categorizes them separately to highlight the conceptual difference of learning in the presence of implicit or explicit information about how tasks are related to each other. Note that these methods may or may not assume access to task indicators during *training*, but they all are unaware of the task indicator during *inference*. Since in either case the agent requires information about task relations, the following discussion omits such distinctions.

### 4.2.1 Regularization

Like in the task-aware setting, a number of task-agnostic techniques aim at avoiding forgetting by penalizing deviations from earlier tasks' solutions. One early method uses a diagonal Gaussian approximation in order to obtain a closed-form update rule based on the variational free energy (Zeno et al., 2018). A later extension of this method handles arbitrary Gaussian distributions by using fixed point equations (Zeno et al., 2021). Another recent technique combines Kronecker-factored EWC (Ritter et al., 2018) with a novel projection method onto the trust region over the posterior of previous tasks (Kao et al., 2021). A distinct approach by Kapoor et al. (2021) trains a variational GP using sparse sets of inducing points per task. Joseph and Balasubramanian (2020) used regularization at a meta level, learning a generative regularized hypernetwork using a variational autoencoder (VAE) to generate parameters for each task based on a task descriptor.

Whereas these methods have operated in the supervised setting, Egorov et al. (2021) recently proposed a VAE model with boosting density approximation for the unsupervised setting.

Functional regularization also applies to the task-agnostic setting for avoiding forgetting. One such method relies on the lottery ticket hypothesis, which states that deep networks with random weight initialization contain much smaller subnetworks that can be trained from the same initialization and reach comparable performance to the original (much larger) network (Frankle and Carbin, 2019). Chen et al. (2021) extended this hypothesis to the lifelong setting by using a pruning and regrowing approach, in combination with functional regularization from unlabeled data from public sources. Another approach combines parameter regularization (specifically, EWC, Kirkpatrick et al., 2017) with functional regularization to avoid forgetting in an approach based on weight and feature calibration (Yin et al., 2021).

### 4.2.2 Replay

Replaying past data stored in memory is another popular mechanism to avoid forgetting in the task-agnostic setting. One recent approach stores data points along with the network's output probabilities, and uses these to functionally regularize the output of the network to stay close to its past predictions (Buzzega et al., 2020). An additional method learns a dual network with different learning rates, training a fast learner to transform a slow learner's output pixel-wise and replaying past data to avoid forgetting (Pham et al., 2021a).

A drastically different technique uses graph learning to discover pairwise similarities between memory and current samples, penalizing forgetting the edges between samples instead of the predictions in order to maintain correlations between samples while permitting significant changes to the network's representations (Tang and Matteson, 2021). An extension to the technique of Gupta et al. (2020a, from the task-aware setting) for lifelong learning via meta-learning trains an additional binary mask that determines which parameters to learn for each task, leading to sparse gradients (von Oswald et al., 2021). Note that this survey categorizes this method as task-agnostic simply because the paper conducted the majority of experiments in that setting.

While these examples have focused on *how* to leverage past examples during training, a related line of work has explored *which* samples to store or replay from the earlier tasks. The method of Aljundi et al. (2019b) stores samples whose parameter gradients are most diverse. Other work proposed to sample from memory the points whose predictions would be affected most negatively by parameter updates without replay (Aljundi et al., 2019a). Chrysakis and Moens (2020) tackled the problem of class imbalance by developing a class-balancing sampling technique to store instances, in combination with a weighted replay strategy. One additional approach determines the optimal points to store in memory using bilevel optimization (Borsos et al., 2020).

Intuitively, it is possible that the training samples are not optimal for lifelong replay (e.g., because they lie far from the decision boundaries). Jin et al. (2021) leveraged this idea by directly modifying the samples in memory via gradient updates to make them more challenging for the learner. Alternatively, one could imagine that storing high-resolution samples might be wasteful, as many features might be superfluous for retaining performance on past tasks. Prior work has exploited this intuition by compressing data in memory via a multilevel VAE that iteratively compresses samples to meet a fixed storage capacity (Caccia et al., 2020).

Like most methods discussed so far, these replay-based approaches have all been used in the vision domain. One exception to this was the work of de Masson d'Autume et al. (2019), which directly applied memory-based parameter adaptation (Sprechmann et al., 2018) with sparse replay to the language domain.

### 4.2.3 Generative replay

Achille et al. (2018) developed a VAE-based approach to lifelong unsupervised learning of disentangled representations, which uses generative replay from the VAE itself to avoid forgetting. A similar technique uses a dynamically expandable mixture of Gaussians to identify when the unsupervised model needs to grow to accommodate new data (Rao et al., 2019). Ayub and Wagner (2021) developed another unsupervised learning method based on neural style transformers (Gatys et al., 2016). Unlike prior methods, this latter approach explicitly stores in memory the autogenerated samples in embedding space, and consolidates them into a centroid-covariance representation to conform to a fixed capacity. As unsupervised approaches, these three methods can seamlessly extend to the supervised setting, as demonstrated in the corresponding manuscripts.

Alternatively, one approach specifically for supervised learning relies on three model components: a set of shared parameters, a dynamic parameter generator for classification, and a data generator (Hu et al., 2019). The data generator generates embeddings both as inputs to the dynamic parameter generator and as replay samples for functional regularization of the shared parameters. A similar approach for supervised learning, inspired by the brain, also replays hidden representations to avoid forgetting (Van de Ven et al., 2020).

### 4.2.4 Expandable models

In the vein of dynamically expandable models, Aljundi et al. (2017) conceived the first task-agnostic method, which trains a separate expert for each task, and automatically routes each data point to the relevant expert at inference time. A distinct method automatically detects distribution shifts during training to meta-learn new components in a mixture of hierarchical Bayesian models (Jerfel et al., 2019). Similarly, the method of Lee et al. (2020) trains a dynamically expandable mixture of experts via variational inference.

### 4.2.5 Closing remarks on task-agnostic lifelong learning

Overall, task-agnostic learning might appear at first glance as an unqualified improvement over task-aware learning. However, most task-agnostic approaches rely on one additional assumption: the task structure must be implicitly embedded in the features of each data point (e.g., one task might be daylight object detection and another nighttime object detection). In some settings, this assumption is not valid, for example if the same data point might correspond to different labels in different tasks (e.g., cat detection and dog detection from images with multiple animals). Figure 4 therefore categorizes these methods as requiring the task structure to be implicitly provided, primarily in the supervised and unsupervised settings, with applications to vision models. To reiterate: in practice, many of the task-aware methods can operate in the task-agnostic setting with minor modifications, and vice versa.

### 4.3 Reusable knowledge

Lifelong approaches discussed so far, although effective in avoiding the problem of catastrophic forgetting, make no substantial effort toward the discovery of reusable knowledge. One could argue that these methods learn the model parameters in such a way that they are reusable across all tasks. However, it is unclear what the reusability of these parameters means, and moreover the architecture design hard-codes how to reuse parameters. This latter issue is a major drawback when attempting to learn tasks with a high degree of variability, as the exact form in which tasks connect to one another is often unknown. One would hope that the algorithm could determine these connections autonomously.

The ELLA framework introduced an alternative formulation based on online dictionary learning (Ruvolo and Eaton, 2013). The elements of the dictionary can be interpreted as reusable models, and task-specific coefficients select how to reuse them. This represents a rudimentary form of functional composition, where each component is a full task model and the new models aggregate the component parameters; in the case of linear models studied in the original paper, this is equivalent to aggregating the component models' outputs.

A few other mechanisms instead autonomously identify which knowledge to transfer across tasks, without explicit compositionality. One approach relies on automatically detecting which layers in a neural network should be specific to a task and which should leverage a shared set of parameters. Existing works have achieved this either via variational inference (Adel et al., 2020) or via expectation maximization (Lee et al., 2021a). Another technique is to identify the most similar tasks by training one model via transfer and another via STL and comparing their validation performances (Ke et al., 2020). Once the agent has identified similar tasks, it uses an attention mechanism to transfer knowledge from only those tasks. An additional algorithm instead avoids explicitly selecting tasks or layers to transfer, and directly meta-learns a set of features that maximize reuse when task-specific parameters mask the shared weights for transfer (Hurtado et al., 2021).

Going back to the illustration of Figure 4, the methods in the previous paragraphs have included applications to vision and have worked only in the supervised setting. As a sole exception, extensions to the work of Ruvolo and Eaton (2013) have applied to RL for robotics tasks, as discussed below in Section 6.

In a distinct line of work, Yoon et al. (2021) developed a knowledge-sharing mechanism for lifelong federated learning, selectively transferring knowledge across clients. In the language domain, Gupta et al. (2020c) achieved lifelong transfer by sharing latent topics.

### 4.4 Additional approaches

While the majority of works on lifelong learning fall into the categories above, some exceptions do not fit this classification. This section briefly describes some recent such efforts.

Javed and White (2019) developed an online meta-learning algorithm that explicitly trains a representation to avoid forgetting. Their work considers a novel problem setting: instead of one lifelong sequence of tasks, the agent faces a pretraining phase, during which it meta-learns the representation over *multiple* "lifelong" sequences of tasks. An extension to this work incorporates a generative classifier (Banayeeanzade et al., 2021). A similar method learns a dual network for gating the outputs of a standard network (Beaulieu et al., 2020).

A different recent problem formulation is that of continual generalized zero-shot learning, which requires the agent to generalize to unseen tasks as well as perform well on all past tasks (Skorokhodov and Elhoseiny, 2021). The authors then presented an algorithm for tackling this new problem via class normalization.

Other works have instead focused on understanding aspects of existing lifelong learning approaches. Mirzadeh et al. (2020) empirically studied the impact of training hyperparameters (dropout, learning rate decay, and mini-batch size) on the width of the obtained local minima—and therefore, on forgetting. This study led to the development of *stable stochastic gradient descent*, a now-popular baseline for benchmarking new lifelong approaches. Another study empirically evaluated the effect of task semantics on catastrophic forgetting, finding that intermediate similarity leads to the highest amount of forgetting (Ramasesh et al., 2021). Lee et al. (2021b) obtained a similar finding for lifelong learning specifically in the teacher-student setting, with additional insight separating task feature similarity and class similarity. A separate work evaluated existing lifelong learning methods on recurrent neural networks, finding that they perform reasonably well and are a solid starting point for developing lifelong methods specific to recurrent architectures (Ehret et al., 2021).

Figure 4 categorizes these last few approaches as lifelong supervised learning methods without any type of composition. Most of the methods either assume implicit information about the structure of the tasks (but no access to a task indicator) or vice versa. The one exception is the work of Skorokhodov and Elhoseiny (2021), which assumes explicit task descriptors that enable zero-shot generalization, which equates to explicit information about the task structure. Similarly, all works considered solely vision applications, with the exception of Ehret et al. (2021), which additionally considered a simple audio application.

## 5 Compositional knowledge

A mostly distinct line of parallel work has explored the learning of compositional knowledge. These methods vary in the information they are given about the structure of the tasks (Figure 6) and the types of compositional representations they learn (Figure 7). This section discusses existing methods for functional composition in the supervised setting, while Section 7 discusses other forms of composition specifically used for RL.

### 5.1 Multitask learning

Most compositional learning methods either learn a set of components given a known structure for how to compose them, or learn the structure for piecing together a given set of components. In the former case, Andreas et al. (2016) proposed to use neural modules as a means for transferring information across VQA tasks. Their method parses the questions in natural language and manually transforms them into a neural architecture. Given this architecture, the agent learns modules for detecting shapes, colors, and spatial relations, and later combines the modules in novel ways to answer unseen questions. In this context, neural modules represent general-purpose, learnable, and composable functions, which permits thinking broadly about functional composition. Consequently, this survey primarily considers learnable components in the form of neural modules.

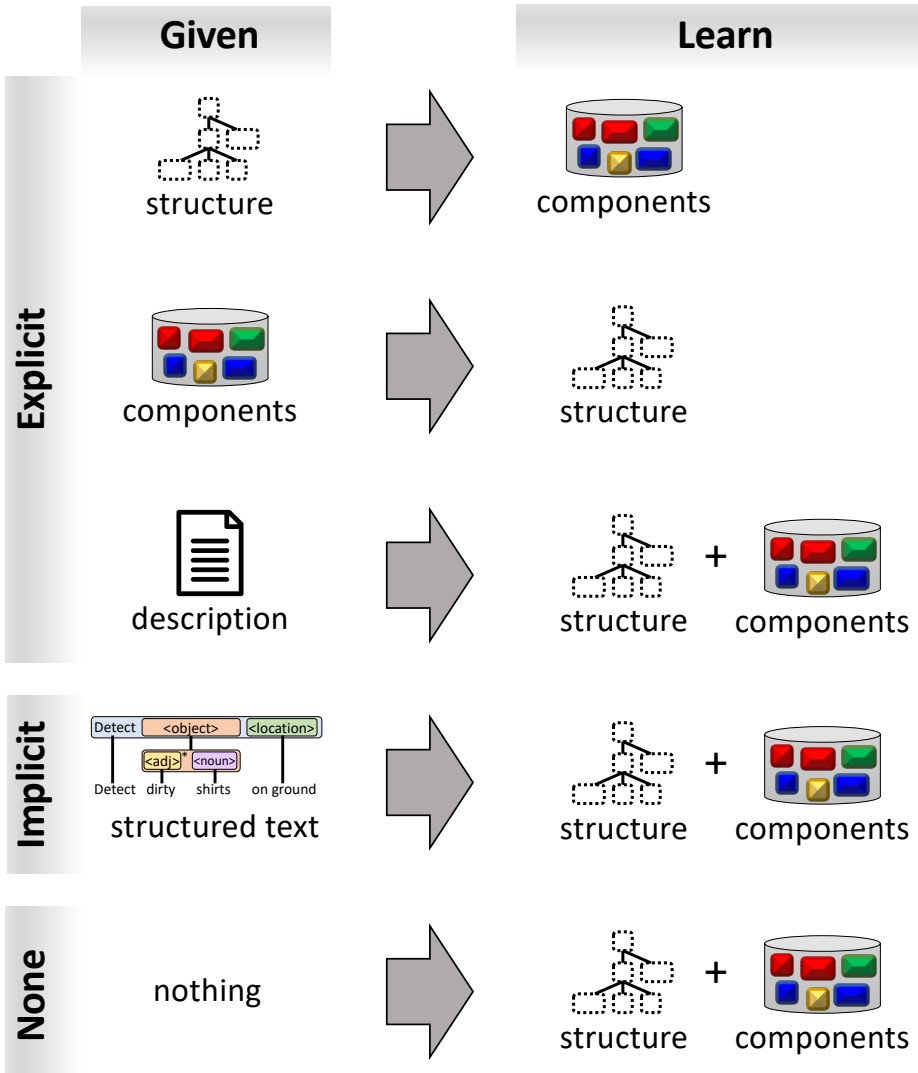

Figure 6: Different forms of compositional learning, depending on what structural information is implicitly or explicitly given to the algorithm and what must be learned.

Following the graph formalism of Section 2.2, each neural module is a node representing one individual subfunction $F_i$, and a sequence of edges over the graph indicates how to compose the modules into an overall solution by passing the output of one module as input to the next. The sequence of edges can be viewed as a selection of which modules to activate for each task and in what order. As an illustrative example of the types of problems that these modular architectures can tackle, consider the example of animal localization and counting from Section 2.2. Figure 8 depicts a compositional solution to the four tasks using modular networks, consisting of modules for detecting cats, detecting dogs, finding largest, and counting.

A related work extended the *neural programmer-interpreter* (NPI, Reed and de Freitas, 2016) to learn an interpreter for the programming language Forth using neural modules as the primitive functions, given manually specified execution traces (Bošnjak et al., 2017). Wu et al. (2021) hypothesized that, in order for neural modules to be composable, they must be invertible, and validated this hypothesis by manually composing invertible modules with themselves and other pretrained modules.

In the latter scenario of a given set of components, Zaremba et al. (2016) learned an RL-based controller to select from among a collection of predefined functions to execute more complex programs. Cai et al. (2017) improved generalization in the NPI framework by incorporating recursion. Another approach based on

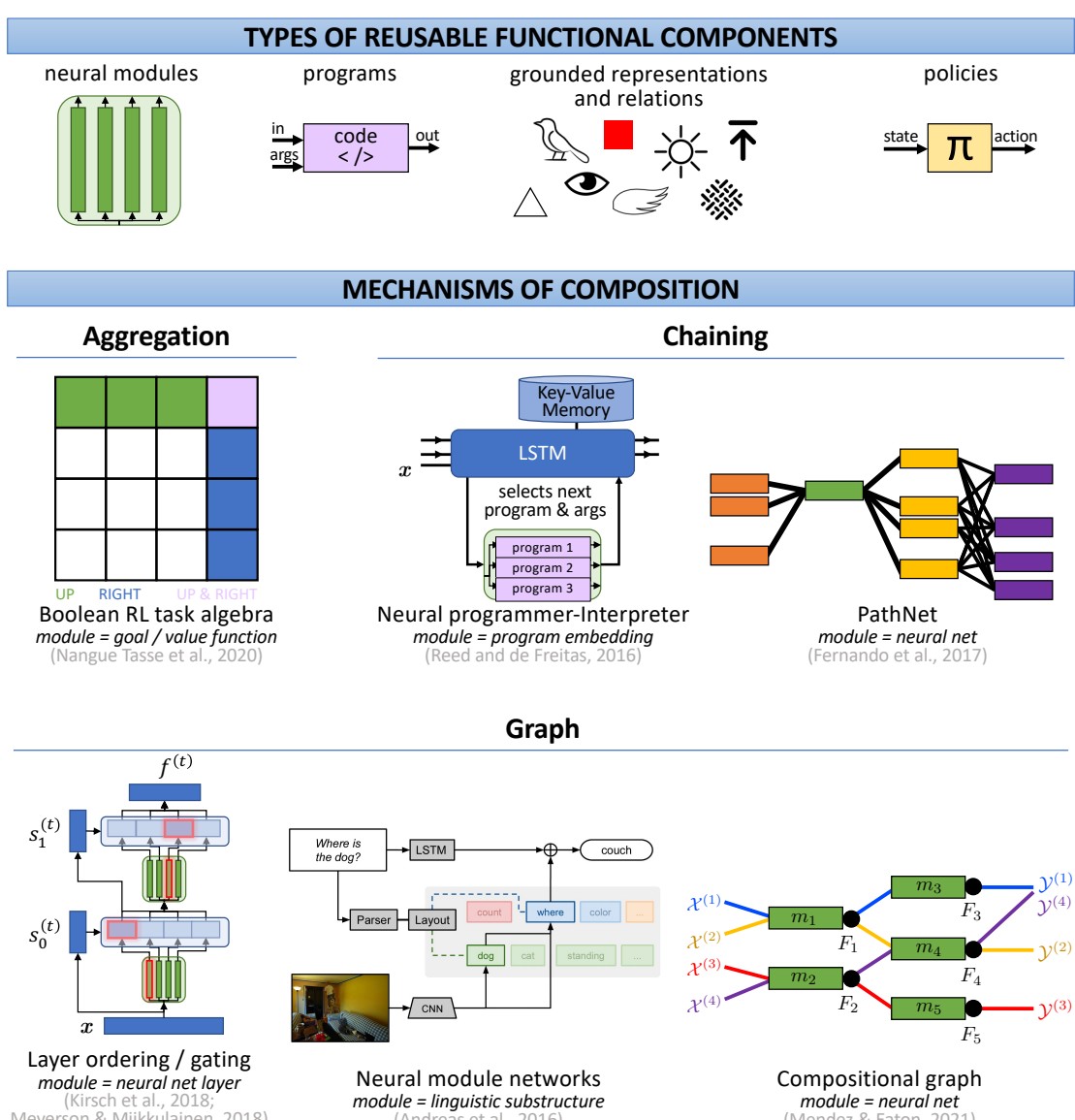

Figure 7: Types of reusable functional components and mechanisms for using those components for MTL and lifelong learning, with illustrative algorithms (neural module networks depiction from Andreas et al. (2016)).

programming languages uses RL for rewarding all semantically correct programs and additionally imposes syntactical correctness directly in the training procedure (Bunel et al., 2018). More advanced RL techniques have tackled the same problem, removing the need for any supervision in the form of annotated execution traces or structures (Pierrot et al., 2019). A separate approach specifically for robot programming tasks uses the application programming interfaces (APIs) of primitive actions to guide the learning (Xu et al., 2018). Recently, similar ideas have achieved compositional generalization by directly learning rules over fixed symbols (Nye et al., 2020) or by providing a curriculum (Chen et al., 2020b). In a related line of work, Saqur and Narasimhan (2020) trained graph neural networks to couple concepts across different modalities (e.g., image and text), keeping the set of possible symbols fixed.

A more interesting case is when the agent knows neither the structure nor the set of components, and must autonomously discover the compositional structure underlying a set of tasks. For example, following Andreas et al. (2016), several approaches for VQA assume that there exists a mapping from the natural language question to the neural structure and automatically learn this mapping (Pahuja et al., 2019). The majority



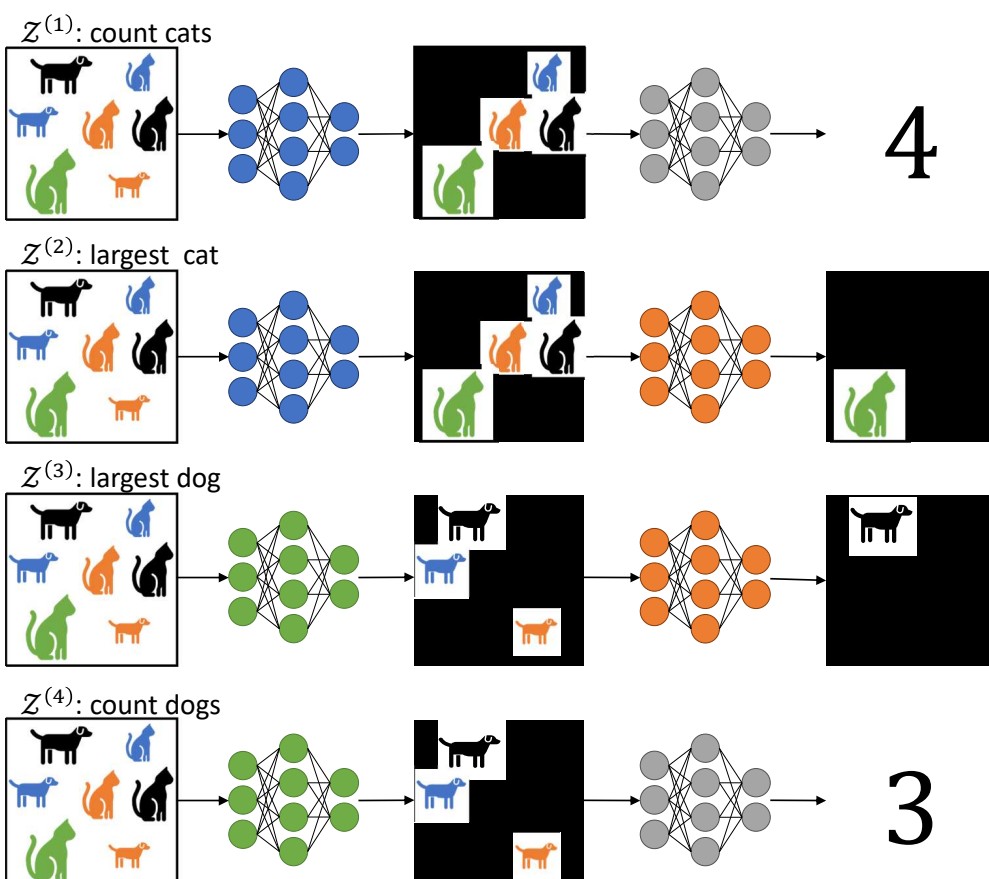

Figure 8: Example of functionally compositional solution using modular neural networks. Each neural module solves one subproblem (e.g., detect dogs). Composing modules by feeding the output of one as the input to the next yields a complete neural network that solves each of the tasks.

of such methods assume access to a set of ground-truth program traces as a supervisory signal for learning the structures. The first such method simply learns a sequence-to-sequence model from text to network architectures in a supervised fashion (Hu et al., 2017). A similar method starts from supervised learning over a small set of annotated programs and then fine-tunes the structure via RL (Johnson et al., 2017). Recent extensions to these ideas have included using probabilistic modules to parse open-domain text (Gupta et al., 2020b) and modulating the weights of convolutional layers with language-guided kernels (Akula et al., 2021).

Figure 4 categorizes the compositional works described so far as supervised MTL methods with explicitly given task structure, either in the form of fixed modules, fixed structures over the modules, or inputs that directly contain the structure (e.g., in natural language). The compositional structure is any arbitrary graph connecting the components, even allowing for components to be reused multiple times in a single task via recursion. Existing works have used these methods in varied application domains: toy programming tasks (Bošnjak et al., 2017; Cai et al., 2017; Bunel et al., 2018; Pierrot et al., 2019), VQA (Andreas et al., 2016; Saqur and Narasimhan, 2020; Pahuja et al., 2019; Hu et al., 2017; Johnson et al., 2017; Gupta et al., 2020b; Akula et al., 2021), natural language (Nye et al., 2020; Chen et al., 2020b), vision (Wu et al., 2021), audio (Wu et al., 2021), and robotics (Xu et al., 2018). As an aside, while most works described so far operate exclusively at the level of abstractions computed by neural modules, others (e.g., those deriving from the NPI framework) operate at a *neuro-symbolic* level. In this case, the outputs of neural modules are mapped to semantically meaningful representations that can be viewed as a form of manually defined interfaces connecting one module to the next, enabling explicit symbolic reasoning.

However, some applications require agents (e.g., service robots) to learn more autonomously, without any kind of supervision on the compositional structures. Several approaches therefore learn this structure directly from the optimization of a cost function. Many such methods assume that the inputs themselves implicitly contain information about their own structure, such as natural language tasks, and therefore use the inputs to determine the structure. One challenge in this setting is that the agent must autonomously discover, in an unsupervised manner, what is the compositional structure that underlies a set of tasks. One approach to this is to train both the structure and the model end-to-end, assuming that the selection over modules is differentiable (i.e., soft module selection, Rahaman et al., 2021). Other approaches instead aim at discovering hard modular models, which increases the difficulty of the optimization process. Methods for tackling this variant of the problem have included using RL (Chang et al., 2019) or expectation maximization (Kirsch et al., 2018) as the optimization tool. These ideas have operated on both vision (Rahaman et al., 2021; Chang et al., 2019) and natural language (Kirsch et al., 2018) tasks.

Other approaches do not assume that there is any information about the structure at all given to the agent, and it must therefore blindly search for it for each task. This often implies that the compositional structure for each task should be fixed across all data points, but often approaches permit reconfiguring the modular structure even *within* a task. On the other hand, much like in the lifelong learning setting without compositional structures, this assumption also implies that the agent requires access to some sort of task indicator. One example of this formulation approximates an arbitrary ordering over a set of neural modules via soft ordering and trains the entire model end-to-end (Meyerson and Miikkulainen, 2018). A related technique decomposes MTL architectures into tensors such that each matrix in the tensor corresponds to a subtask, using hypermodules (akin to hypernetworks) to generate local tensors (Meyerson and Miikkulainen, 2019). Another example assumes a hard module selection, and trains the modules via meta-learning so that they are able to quickly find solutions to new, unseen tasks (Alet et al., 2018). An extension of this method learns with graph neural networks (Alet et al., 2019), and a simplified version discovers whether modules should be task-specific or shared via Bayesian shrinkage (Chen et al., 2020c). The approach of Rosenbaum et al. (2018) also learns a hard module selection, but uses RL to select the modules to use for each data point and task. One of the advantages of keeping the structural configuration fixed for each task (instead of input-dependent) is that the reduced flexibility protects the model from overfitting. This has enabled applying these latter methods to domains with smaller data sets than are typically available in language domains (Meyerson and Miikkulainen, 2019; Chen et al., 2020c), such as vision (Rosenbaum et al., 2018; Meyerson and Miikkulainen, 2018; 2019) and robotics (Alet et al., 2018; 2019).

Rosenbaum et al. (2019) discussed the challenges of optimizing modular architectures via an extensive evaluation with and without task indicators on vision and language tasks.

## 5.2 Lifelong learning

All compositional methods described so far assume that the agent has access to a batch of tasks for MTL, enabling it to evaluate numerous combinations of components and structures on all tasks simultaneously. In a more realistic setting, the agent faces a sequence of tasks in a lifelong learning fashion. Most work in this line has assumed that the agent can fully learn each component by training on a single task, and then

reuse the learned module for other tasks. One example is the NPI, which assumes that the agent receives supervised module configurations for a set of tasks and can use this signal to learn a mapping from inputs to module configurations (Reed and de Freitas, 2016). Extensions to the NPI have operated in the MTL setting and were described in the previous section. Other methods do not assume that there is any information in the input about the task structure, and therefore must search for the structure for every new task. Fernando et al. (2017) trained a set of neural modules and chose the paths for each new task using an evolutionary search strategy, applying this technique to both supervised and RL. These methods maintain a constant number of modules and keep the weights of those modules fixed after training them on a single task, limiting their applicability to a small number of tasks. To alleviate these issues, other methods progressively add new modules, keeping existing modules fixed (Li et al., 2019). This is akin to capacity expansion methods from Section 4. Some such approaches introduce heuristics for searching over the space of possible module configurations upon encountering a new task to improve efficiency, for example using programming languages techniques (Valkov et al., 2018) or data-driven heuristics (Veniat et al., 2021). In the language domain, Kim et al. (2019) developed an approach that progressively grows a modular architecture for solving a VQA task by providing a curriculum that imposes which module solves which subtask, keeping old modules fixed.

Unfortunately, this solution of keeping old modules fixed is infeasible in many real-world scenarios in which the agent has access to little data for each of the tasks, which would render these modules highly suboptimal. Therefore, other methods have permitted further updates to the model parameters. One early example, based on programming languages, simply assumed that future updates would not be harmful to previous tasks (Gaunt et al., 2017). This limited the applicability of the method to very simplistic settings. Rajasegaran et al. (2019) proposed a more complete approach that combines regularization and replay to avoid catastrophic forgetting, but requires expensively storing and training multiple models for each task to select the best one before adapting the existing parameters. Another approach routes each data point through a different path in the network, restricting updates to the path via EWC regularization if the new data point is different from past points routed through the same path (Chen et al., 2020a). However, this latter approach heavily biases the obtained solution toward the first task, and does not permit the addition of new modules over time.

A recent, more complete framework uses multiple stages for initializing, reusing, adapting, and spawning components, without access to large batches of simultaneous tasks or expensive training of multiple parallel models (Mendez and Eaton, 2021). The authors further demonstrated the compatibility of compositional methods with replay, regularization, and capacity expansion approaches for lifelong training. Qin et al. (2021) developed a similar supervised learning approach that automatically grows and updates modules for each new task using an RL-based controller. While the approach of Mendez and Eaton (2021) sidesteps forgetting in the structure over modules by making it task-specific, the controller of Qin et al. (2021) is susceptible to catastrophic forgetting. Ostapenko et al. (2021) developed a local per-module selector that estimates whether each sample is in-distribution for the given module, and chooses the module with the highest value. This mechanism lets their method operate in the task-agnostic setting and limits forgetting to local, per-module parameters. While this addresses a large part of the problem of forgetting in the module-selection stage, it enables earlier tasks to select new modules that are likely to malfunction in the presence of old data they did not train on. Notably, this method demonstrated the ability of existing modules to combine in novel ways to solve unseen tasks, exhibiting for the first time compositional generalization in the lifelong learning setting.

Most approaches described in this section assume an arbitrary graph structure over the components, and learn to construct paths through this graph. In the case of neural modules, this means that each module can be used as input to any other module, or equivalently that modules can be chosen at any depth of the network. Some exceptions in the lifelong setting impose a chaining structure by restricting certain modules to be eligible only at certain depths. Note that both of these choices contemplate an exponential number (in the network's depth) of possible configurations. However, the chaining approach does simplify the problem of learning modules, since it reduces the space of possible inputs and outputs that each module must accept and generate, respectively.

## 5.3  Nonmodular compositional works

While modular neural architectures have become popular for addressing compositional problems, they are not the only solution. A number of works have dealt with the problem of compositional generalization to unseen

textual tasks, where an agent may have learned the concepts of "walk", "twice", and "turn left" separately, and later be required to parse an instruction like "walk twice and turn left" (Lake and Baroni, 2018).

One approach uses meta-learning to explicitly optimize the agent to reason compositionally by generalizing to unseen combinations of language instructions (Lake, 2019). Another method, inspired by the emergence of compositionality in human language, uses iterated learning on neural networks to compositionally generalize (Ren et al., 2020). Gordon et al. (2020) equated language composition to equivariance on permutations over group actions, and designed an architecture that maintains such equivariances. A similar work imposed invariance to partial permutations on a language understanding system (Guo et al., 2020a). Another recent technique incorporates a memory of automatically extracted analytical expressions and uses those to compositionally generalize (Liu et al., 2020). A distinct approach by Akyürek et al. (2021) uses data augmentation to specifically target compositionality, combining prototypes of a generative model into multiprototype samples.

One method in this space operates in the lifelong setting, where the vocabulary grows over time (Li et al., 2020b). The agent separates the semantics and syntax of inputs, keeping the syntax for previously learned semantics parameters fixed and learning additional semantics parameters for each extension of the vocabulary.

The literature on visual object detection has also studied the idea of compositional generalization, under the vein of attribute-based zero-shot classification. At a high level, objects in images contain annotations not only of their class label but also of a set of attributes (e.g., color, shape, texture), and the learning system seeks to detect unseen classes based on their attributes. This requires the agent to learn both the semantics of the attributes and how to combine them (Huynh and Elhamifar, 2020; Atzmon et al., 2020; Ruis et al., 2021).

Other approaches compose attributes to generate images. One such method learns one energy-based model per attribute that can later be combined with other attributes in novel combinations—for example, to generate a smiling man from the attributes "smiling" and "man" (Du et al., 2020). The approach of Aksan et al. (2020) learns embeddings of manual drawings that can be composed into complex figures like flowcharts. Similarly, the mechanism of Arad Hudson and Zitnick (2021) uses GANs with structural priors to generate scenes by composing multiple objects. A different method instead relies on large-scale data sets to compositionally generate images without any explicit notion of composition during training (Ramesh et al., 2021; 2022). Another approach that leverages large-scale data uses pretrained language models in conjunction with external tools (such as a calculator and a calendar) to solve compositional tasks (Schick et al., 2023).

While related, this line of work is farther from the notion of composition studied in this review, and so a comprehensive overview is outside of the scope of this discussion. Moreover, even though these methods enable generalizing compositionally, they lack the explicit modularity that would enable the addition of new components over time or the improvement of specific components, as required for true lifelong learning.

## 5.4 Understanding and measuring composition

A recent line of work has sought to understand various aspects of compositionality. An initial study quantified the compositionality of a model as the ability to approximate its output on compositional inputs by combining representational primitives (Andreas, 2019). Evaluating a set of models under this measure, the author found a correlation (albeit small) between compositionality and generalization on vision and language tasks. A similar study found that the same definition of compositionality is related to zero-shot generalization on vision tasks (Sylvain et al., 2020). Schott et al. (2022) found that representation learning approaches do not compositionally generalize to new combinations of known factors of variation. D'Amario et al. (2021) showed that (manually defined) explicitly modular neural architectures improve compositional generalization in VQA tasks. Somewhat contradictorily, Agarwala et al. (2021) found theoretically and empirically that a single monolithic network is capable of learning multiple highly varied tasks. However, this ability requires an appropriate encoding of the tasks that separates them into clusters. One work used a similar intuition to develop a mechanism to compute a description of the execution trace of a modular architecture based on random matrix projections onto separate regions of an embedding space (Ghazi et al., 2019). Given the apparent importance of modularity and compositionality, Csordás et al. (2021) studied two properties of neural networks without modular architectures: whether they automatically learn specialized modules, and whether they reuse those modules. While they found that neural networks indeed automatically learn highly specialized modules, they do not automatically reuse those, thereby inhibiting compositional generalization.

### 5.5 A caveat on compositional learning methods

The methods described in this section for discovering compositional knowledge rely on the assumption that the learning problems faced by the agent are related via some underlying compositional structure, as described in Section 2.2. Incorporating this assumption into learning algorithms constitutes a form of bias, which may hinder performance in instances where the problems are not compositionally related as expected. This is in contrast with the more general lifelong learning formulation of Equation 1, which makes no assumption about the relations between tasks. However, note that model sharing across tasks lies on a spectrum, where full sharing (Figure 3b) and no sharing (Figure 3a)—the most common approaches to the lifelong learning problem—form the two extrema, and compositionality (Figure 3c) is somewhere in between. Developing algorithms that can autonomously recover where the optimal solution to a set (or sequence) of learning tasks lies on this spectrum remains an open challenge.

## 6 Lifelong reinforcement learning

The techniques discussed so far primarily deal with supervised learning tasks. The number of approaches that operate in the lifelong RL setting is substantially smaller. The following paragraphs describe some of the existing methods for lifelong RL in relation to the functional compositionality discussed in this article.

Much like in the supervised setting, the majority of lifelong RL approaches rely on monolithic or nonmodular architectures, which as discussed in Section 4 inhibits the discovery of self-contained and reusable knowledge. These methods mainly use regularization techniques for avoiding forgetting, in a manner that is conceptually equivalent to the supervised methods of Sections 4.1.1 and 4.2.1. A prominent example is EWC (Kirkpatrick et al., 2017), a supervised method that imposes a quadratic penalty for deviating from earlier tasks' parameters, which has been directly applied to RL. One challenge in training RL models via EWC is that the vast exploration typically required to learn new RL tasks might be catastrophically damaging to the knowledge stored in the shared parameters. Consequently, as an alternative approach, *progress & compress* first trains an auxiliary model for each new task and subsequently discards it and distills any new knowledge into the single shared model via an approximate version of EWC (Schwarz et al., 2018). While these methods in principle can handle task-agnostic settings, assuming that the input contains implicit cues about the task structure, in practice evaluations have tested them most often in the task-aware setting, typically in vision-based tasks (e.g., Atari games, Bellemare et al., 2013). Moreover, these works have dealt with limited lifelong settings, with relatively short sequences of tasks and permitting the agent to revisit earlier tasks several times for additional training experience. Even in these simplified settings, these methods have failed to achieve substantial transfer over an agent trained independently on each task, without any transfer. Kaplanis et al. (2019) trained a similar monolithic architecture in the task-agnostic setting, including for tasks with continuous distribution shifts. Their approach regularizes the KL divergence of the policy to lie close to itself at different timescales, and was evaluated on simulated continuous control tasks.

Other approaches store experiences for future replay. Compared to the supervised methods of Sections 4.1.2 and 4.2.2, the use of experience replay to retain performance on earlier RL tasks requires a number of special considerations. For example, the data collected over the agent's training on each individual task is nonstationary, since the behavior of the agent changes over time. Isele and Cosgun (2018) proposed various techniques for selectively storing replay examples and compared their impact on performance. Another challenge is that, as the agent modifies the policy for earlier tasks, the distribution of the data stored for them no longer matches the distribution imposed by the agent's policy. Rolnick et al. (2019) proposed using an importance sampling mechanism for limiting the effects of this distributional shift. While the former example considered mostly grid-world-style tasks in the task-aware setting, where the input contains no information about the task relations, the latter considered vision-based tasks in the task-agnostic setting, under the assumption that the observation space for each task contains sufficient information for distinguishing it from others. However, the challenges of replay in RL have limited the applicability of these methods to short sequences of two or three tasks, still with the ability to revisit previous tasks. Mendez et al. (2022b) established a connection between these issues and off-line RL (Levine et al., 2020), and leveraged it to develop a replay mechanism that operates on sequences of tens of tasks without revisits. Berseth et al. (2022) also used replay in longer sequences of RL tasks in conjunction with off-line meta-RL to achieve lifelong transfer.

While most lifelong RL works have considered the use of a single monolithic structure for learning a sequence of tasks, some classical examples have instead followed the ELLA framework of Ruvolo and Eaton (2013) to devise similar RL variants. PG-ELLA follows the dictionary-learning mechanics of ELLA, but replaces the supervised models that form ELLA's dictionary by policy factors (Bou Ammar et al., 2014). An extension of this approach supports cross-domain transfer by projecting the dictionary onto domain-specific policy spaces (Bou Ammar et al., 2015), and another extension leverages task descriptors to achieve zero-shot transfer to unseen tasks (Isele et al., 2016; Rostami et al., 2020). Zhao et al. (2017) followed a similar dictionary-learning formulation for deep networks in the batch MTL setting, replacing all matrix operations with equivalent tensor operations. Like ELLA in the supervised setting (Section 4.3), this represents a rudimentary form of aggregated composition per the categorization of Figure 4. The primary challenge that ELLA-based approaches face is that the dictionary-learning technique requires first discovering a policy for each task in isolation (i.e., ignoring any information from other tasks) to determine similarity to previous policies, before factoring the parameters to improve performance via transfer. The downside is that the agent does not benefit from prior experience during initial exploration, which is critical for data-efficient training in lifelong RL. While these methods target continuous control tasks, their evaluations have considered the interleaved MTL setting, where the agent revisits tasks multiple times before evaluation.

Mendez et al. (2020) developed LPG-FTW, a mechanism that uses multiple models like PG-ELLA variants, but learns these models directly via RL training like single-model methods. This enables the method to be flexible and handle highly varied tasks while also benefiting from prior information during the learning process, thus accelerating the training. A similar approach in the context of model-based RL represents the dynamics of the tasks via an aggregation of supervised models (Nagabandi et al., 2019).

Other approaches instead use an entirely separate model for each task. One such method leverages shared knowledge in the form of a metamodel that informs exploration strategies to task-specific models, resulting in linear growth of the model parameters (Garcia and Thomas, 2019). Another popular example shares knowledge via lateral connections in a deep network, resulting in quadratic growth of the model parameters (Rusu et al., 2016). Both of these approaches are infeasible in the presence of large numbers of tasks.

A separate line of lifelong RL work has departed completely from the notion of tasks and has instead learned information about the environment in a self-guided way. The seminal approach in this area is *Horde*, which learns a collection of general value functions for a variety of signals and uses those as a knowledge representation of the environment (Sutton et al., 2011). A more recent approach learns latent skills that enable the agent to reset itself in the environment in a way that encourages exploration (Xu et al., 2020).

## 7 Compositional reinforcement learning

The most common form of composition studied in RL has been temporal composition. One influential work in this area is the *options* framework of Sutton et al. (1999). At a high level, options represent temporally extended courses of action, which can be thought of as skills. Once the agent has determined a suitable set of options, it can then learn a higher-level policy directly over the options. In the language used so far, each option is a module or component, and the high-level policy is the structural configuration over modules.

Traditional work in temporal composition has assumed that the environment provides the structure a priori as a fixed set of options (or information about how to learn each option, such as subgoal rewards). For example, the approach of Lee et al. (2019) learns a policy for transitioning from one skill to the next, given a set of pretrained skills. However, other methods automatically discover both the modules and the configuration over them. *Option-critic* extends actor-critic methods to handle option discovery via an adaptation of the policy gradient theorem (Bacon et al., 2017). Another approach uses a large off-line data set to automatically extract skills and later meta-learns policies on top of these extracted skills (Nam et al., 2022). Recent work has developed mechanisms for skill chaining other than an explicit high-level policy, such as additively combining abstract skill embeddings (Devin et al., 2019) or multiplicatively combining policies (Peng et al., 2019).

The high expressive power of a policy over options enables learning arbitrary graphs over the modules according to the categorization of Figure 4. However, these approaches have primarily been limited to toy applications, with some exceptions considering simple visual-based or continuous control tasks.

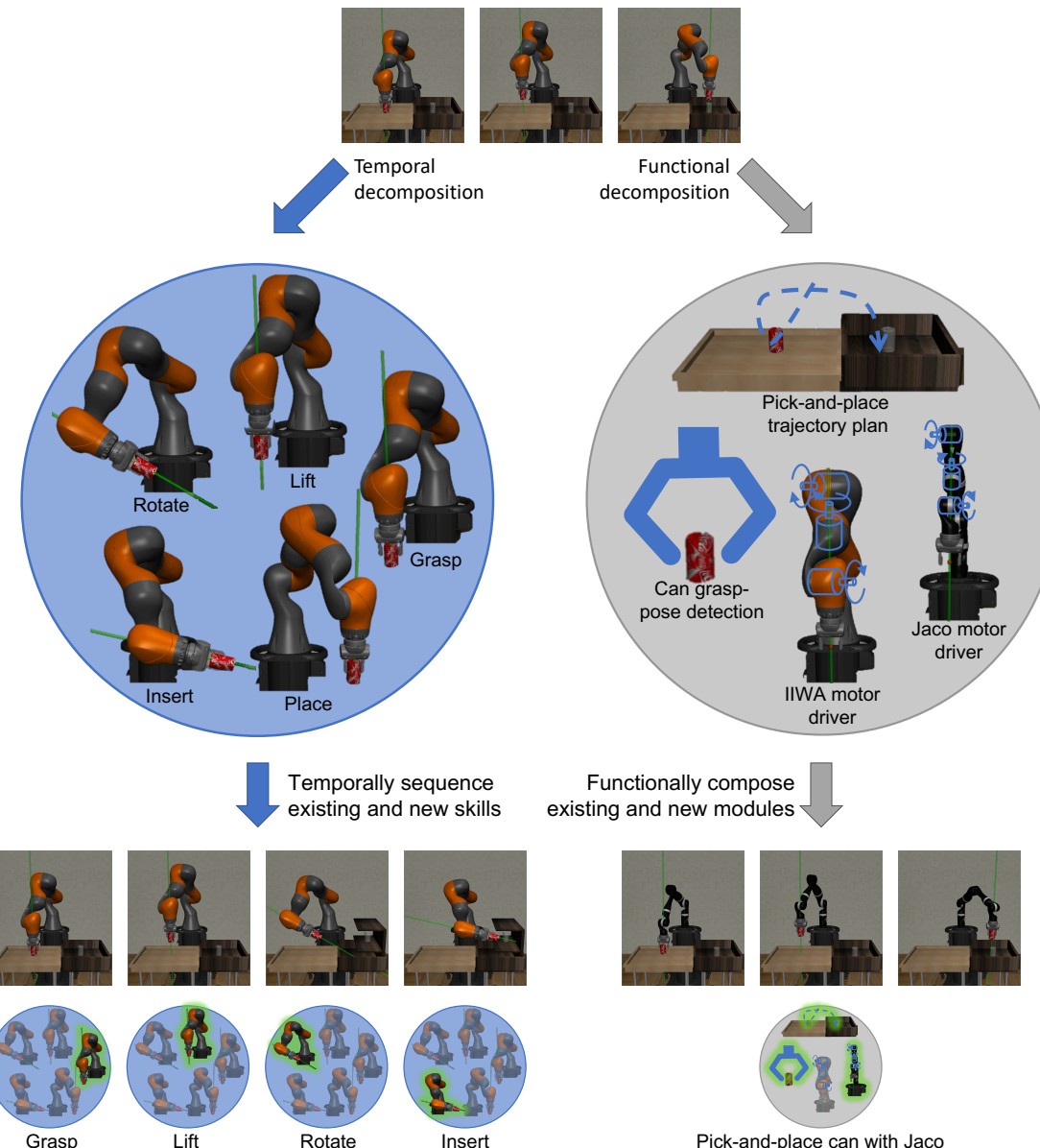

Figure 9: Functional vs. temporal composition of RL policies. The policy to pick-and-place a can with an IIWA arm is decomposed along a temporal or a functional axis. Temporally extended actions correspond to skills, such as grasp, lift, or place, which are transferable to a place-on-shelf task with the IIWA arm. Functional components correspond to processing stages, such as grasp-pose detection, trajectory planning, or motor control, which are transferable to a pick-and-place task with a different arm, like a Jaco. Temporal modules are active one at a time, while multiple functional components activate simultaneously.

Crucially, the problem considered in this article differs in that the functional composition occurs at every time step, as opposed to the temporal chaining of options. As illustrated in Figure 9, these two dimensions are orthogonal, and both capture real-world settings in which composition would greatly benefit the learning process of artificial agents. More concretely, the primary conceptual difference between hierarchical RL and functionally compositional RL is that hierarchical RL considers the composition of sequences of actions *in time*, whereas functionally compositional RL considers the composition *of functions* that, when combined, form a full policy. In particular, for a given compositional task, the agent uses all functions that make up its modular policy at every time step to determine the action to take (given the current state).

Going back to the example of robot programming from Section 2.2, modules in the compositional RL formulation might correspond to sensor processing units, path planners, or robot motor drivers. In programming, at every time step, the sensory input passes through modules in some preprogrammed sequential order, which finally outputs the motor torques to actuate the robot. Similarly, in compositional RL the state observation passes through the different modules, used in combination to execute the agent's actions. The choice to focus on this form of functional composition permits us to analyze supervised and RL methods under the common lens of compositional problem graphs described in Section 2.2.

Hierarchical RL takes a complementary approach. Instead, each "module" (e.g., an option) is a self-contained policy that receives as input the state observation and outputs an action. Each of these options operates in the environment, for example to reach a particular subgoal state. Upon termination of an option, the agent selects a different option to execute, starting from the state reached by the previous option. In contrast, the compositional RL framework assumes that the agent uses a single policy to solve a complete task.

An integrated approach is possible that decomposes the problem along both a functional axis and a temporal axis. This would enable selecting a different functionally modular policy at different stages of solving a task, simplifying the amount of information that each module should encode. Conversely, it would enable options to be made up of functionally modular components, simplifying the form of the options themselves and enabling reuse *across* options. Research in this direction could drastically improve RL data efficiency.

With this in mind, note that the discussion of works on skill discovery, which is a vast literature on its own, is by no means comprehensive. We refer the reader to separate surveys for a deeper study of this line of work (Barto and Mahadevan, 2003; Al-Emran, 2015; Pateria et al., 2021).

Other forms of hierarchical RL have considered learning state abstractions that enable the agent to more easily solve tasks (Dayan and Hinton, 1993; Dietterich, 2000; Vezhnevets et al., 2017). While these are also related, they have mainly focused on a two-layer abstraction. This represents a simple form of composition where the agent executes actions based on a learned abstracted state. Instead, general functional composition considers arbitrarily many layers of abstraction that help the learning of both state and action representations.

Most works on both temporal composition and state abstractions have been in the STL setting, where the agent must simultaneously learn to solve the individual task and learn to decompose its knowledge into suitable components. In practice, this has implied that there is not much benefit to learning such a decomposition, since the learning itself becomes more costly. However, other investigations have considered learning compositional structures for multiple tasks, in particular in the lifelong setting. Brunskill and Li (2014) developed a theoretical framework that automatically discovers options and policies over options throughout a sequence of tasks. A more practical approach trains each option separately on a subtask, and later reuses these options for learning subsequent tasks (Tessler et al., 2017). A recent model-based approach learns skills in an off-line phase that subsequently enable the agent to learn in a nonstationary lifetime without explicit tasks (Lu et al., 2021). Other work studied state abstractions from a theoretical perspective in the lifelong setting (Abel et al., 2018). When learning such compositional structures in a lifelong setting, the agent amortizes the cost of decomposing knowledge over the multiple tasks, yielding substantial benefits when the components capture knowledge that is useful in the future.

Another form of composition studied in the RL literature has been to learn behaviors that solve different objectives and compose those behaviors to achieve combined objectives. Todorov (2009) showed that the linear composition of value functions is optimal in the case of linearly solvable Markov decision processes (MDPs). A similar result showed that successor features can be combined to solve this type of combined objectives (Barreto et al., 2018). A recent work leveraged this result to develop a simple algorithm for constructing a set of policies that can be combined into optimal policies for all possible tasks expressible with a particular set of successor features (Alver and Precup, 2022). One common terminology for discussing how this process combines objectives is logical composition. Intuitively, if an agent has learned to solve objective $A$ and objective $B$ separately, it can then combine its behaviors to solve $A$ AND $B$ or $A$ OR $B$. This intuition has driven theoretical and empirical results in the setting of entropy-regularized RL (Haarnoja et al., 2018; Van Niekerk et al., 2019). One approach in this setting explicitly modularizes the inputs to a neural network to handle each of the different goals, aided by multi-hot indicators of the active goals (Colas et al., 2019). Nangue Tasse et al. (2020; 2022) later formalized the intuition of logical composition in the lifelong

setting. Other recent work developed this idea of composing multiple simultaneous behaviors specifically for robotic control (Cheng et al., 2021; Li et al., 2021a; Bylard et al., 2021). A related line of work designed a formal language for specifying logically compositional tasks (Jothimurugan et al., 2019), and later used a similar language to learn hierarchical policies (Jothimurugan et al., 2021). A closely related work used differentiable program search to learn compositional policies (Qiu and Zhu, 2022). These compositional approaches require a specification of compositional objectives. Another related vein has sought to decompose the reward into such components, and learn separate policies for each component that can later be combined. These works have decomposed the reward manually (Van Seijen et al., 2017) or automatically (Lin et al., 2019; 2020). In practice, these logic-based approaches typically combine behaviors via simple aggregation of value functions (e.g., weighted combination or addition), which limits their applicability to components that represent solutions to entire RL problems. In contrast, the more general functional composition separates each policy itself into components, such that these components combine to form full policies.

A handful of works have considered functional composition in RL with modular neural networks, resulting in chaining or full graph compositional solutions. A first method handles a setting where each task is a combination of one robot and one task objective (Devin et al., 2017). Given prior knowledge of this compositional structure, the authors manually crafted chained modular architectures and trained the agent to learn the parameters of the neural modules. Other works have instead assumed no knowledge of the task structure and learned them autonomously, under the assumption that the inputs contain implicit cues of what distinguishes the modular structure of one task from another. In this line, one recent technique learns *recurrent independent mechanisms* by encouraging modules to become independent via a competition procedure, and combines the modules in general graph structures (Mittal et al., 2020; Goyal et al., 2021; 2022). These methods were originally developed for the supervised setting and were directly applied to RL, and have primarily operated in the STL setting. Another closely related method also automatically learns a mapping from inputs to modular structures in the MTL setting, with applications to noncompositional robotic manipulation (Yang et al., 2020). More recently, one approach was developed that tackles explicitly compositional RL tasks in a lifelong setting, with applications to robotic manipulation (Mendez et al., 2022b).

Compositionality has a long history in RL, given the promise that learning smaller, self-contained policies might make RL of complex tasks feasible. This has led to a wide diversity of ways to define composition. For completeness, this paragraph discusses other recent approaches to compositionality that have received less attention and bear less connection to the work presented here. As one example, Pathak et al. (2019) sought to decompose policies via a graph neural network such that each node in the graph corresponds to a link in a modular robot. A later version of this work extended the method by considering a setting where all links are morphologically equivalent in terms of their size and motor, and ensuring that all modules learn the same policy (Huang et al., 2020). Others have learned object-centric embeddings in order to generalize to environments with different object configurations (Li et al., 2020a; Mu et al., 2020). Li et al. (2021b) developed an approach related to skill discovery, but instead of combining skills, the agent learns to solve progressively harder tasks by truncating demonstrated trajectories in an imitation learning setting, such that the starting state leads to a task solvable by the current agent.

The understanding of the modularity of RL agents at a fundamental level has received very little attention. Perhaps the one exception has been the recent work of Chang et al. (2021), which studied the modularity of credit assignment as the ability of an algorithm to learn mechanisms for choosing actions that can be modified independently of the mechanisms for choosing other actions. The conclusion of this study was that some single-step temporal difference methods are modular, but policy gradient methods are not.

### 7.1 Benchmarking compositional reinforcement learning

Large-scale, standardized benchmarks have been key to the acceleration of deep learning research (e.g., ImageNet, Deng et al., 2009). Inspired by this, multiple attempts have sought to construct equivalent benchmarks for deep RL, leading to popular evaluation domains in both discrete- (Bellemare et al., 2013; Vinyals et al., 2017) and continuous-action (Brockman et al., 2016; Tunyasuvunakool et al., 2020) settings.

While these benchmarks have promoted deep RL advancements, they are restricted to STL—that is, they design each task to be learned in isolation. Consequently, work in multitask and lifelong RL has resorted to

ad hoc evaluation settings, slowing down progress. Recent efforts have sought to close this gap by creating evaluation domains with multiple tasks that share a common structure that is (hopefully) transferable across the tasks. One example varied dynamical system parameters of continuous control tasks (e.g., gravity) to create multiple related tasks (Henderson et al., 2017). Other work created a grid world evaluation domain with tasks of progressive difficulty (Chevalier-Boisvert et al., 2019). In the continual learning setting, a recent benchmark evaluates approaches in a multiagent coordination setting (Nekoei et al., 2021). Specifically in the context of robotics, recent works have created large sets of tasks for evaluating MTL, lifelong learning, and meta-learning algorithms (Yu et al., 2019; James et al., 2020; Wołczyk et al., 2021).

Despite this recent progress, it remains unclear exactly what an agent can transfer between tasks in these benchmarks, and so existing algorithms are typically limited to transferring neural network parameters in the hopes that they discover reusable information. Unfortunately, typical evaluations of compositionality use such standard benchmarks in both the supervised and RL settings. While this enables fair performance comparisons, it fails to give insight into the agent's ability to find meaningful compositional structures. Some notable exceptions exist for evaluating compositional generalization in supervised learning (Bahdanau et al., 2018; Lake and Baroni, 2018; Sinha et al., 2020; Keysers et al., 2020).

Recently, Mendez et al. (2022a) developed an RL benchmark for functional compositional generalization in a robotic manipulation setting with varied robot arms, and Gur et al. (2021) developed a complementary benchmark for temporal composition. Another related work procedurally created robotics tasks by varying dynamical parameters to study causality in RL (Ahmed et al., 2021), considering a single robot arm dealing with continuous variations in the physical properties of objects. These evaluation domains represent initial steps toward objectively quantifying the compositional capabilities of RL approaches. Much work remains to be done to develop both algorithms that tackle these difficult benchmarks, and new benchmarks and metrics that that more generally evaluate and shed light on compositionality.

## 8 Summary and directions for future work

This article reviewed the state of prior research on the topics of lifelong learning and compositional learning, and categorized them along six dimensions. In summary, lifelong or continual learning has primarily focused on the problem of catastrophic forgetting in the supervised setting, but has mostly overlooked how to obtain knowledge that can be reusable for future tasks. On the other hand, compositional learning has developed methods for obtaining reusable knowledge, but has done so in the simpler case of MTL, where the agent trains on all tasks simultaneously. Few works have combined these two lines of work by developing algorithms that discover reusable compositional knowledge in a lifelong setting. Similarly, not much effort has been placed in porting lifelong learning techniques to the RL setting, and methods in this line have had shortcomings that have prevented their application to complex and diverse sequences of tasks. In particular, the form of functional composition discussed here has been severely understudied in the RL literature.

Despite the relative lack of attention that lifelong compositional learning has received in the literature, it is the authors' perspective that this joint problem space holds promise for tremendously fruitful advances. The main arguments that support this claim are outlined below:

- **Knowledge reuse.** While the field of lifelong learning has focused primarily on the problem of catastrophic forgetting in recent years, lifelong learning really is the study of the discovery and effective utilization of reusable knowledge. An agent tasked with efficiently learning over a stream of non-$i.i.d.$ data—presented as a sequence of tasks or otherwise—should leverage any information gathered earlier that is related to any aspect of the problem in the current state of the distribution. This form of reuse would permit far more data-efficient learning and broader generalization. Compositionality is a natural, extremely general mechanism for reusing knowledge, by composing chunks of knowledge in various combinations to solve novel problems over time. Moreover, thanks to combinatorial explosion, compositionality enables reuse across a multitude of problems at scale.

- **Modular knowledge updates.** Explicitly modular architectures offer another clear advantage over nonmodular ones: their potential for updating individual elements of the model to account

for new information. This is especially relevant for avoiding catastrophic forgetting, since at any point in time the agent would only need to modify knowledge explicitly used to solve the current task or data point, which should represent only a small portion of the overall knowledge the agent has accumulated over its lifetime. This property is also advantageous for simplifying training, since limiting gradient updates to only relevant modules reduces the difficulty of credit assignment.

- **Early evidence from existing approaches.** As a more pragmatic argument, the few existing compositional approaches to lifelong learning have exhibited drastic performance improvements over competing techniques, especially in settings with highly diverse tasks. As examples, Mendez et al. (2020) demonstrated an improvement of 82.5% over noncompositional methods, and Ostapenko et al. (2021) demonstrated zero-shot compositional generalization to novel task combinations.

- **Advances in (non-lifelong) compositional learning.** Some years ago, the problem of automatically learning compositional neural networks seemed hopeless, which likely (understandably) drove lifelong learning researchers away from these types of models. However, the landscape has changed significantly in recent years, with many novel mechanisms enabling the learning of high-quality modular architectures. In particular, such approaches vary in the types of assumptions they make about the information the designers must provide to the agent about the relations among tasks. For example, Andreas et al. (2016) assume that the structure in which modules are combined must be given (explicit information), Zaremba et al. (2016) assume that the modules themselves are given and the agent must discover how to combine them (explicit information), Hu et al. (2017) assume textual descriptions directly encode how to combine modules (explicit information), Kirsch et al. (2018) assume the input contains cues of how a solution might be encoded (implicit information), and Alet et al. (2018) assume there is no information at all in the input and it must be discovered via search (no information). As one would expect, these choices impose significant trade-offs between the difficulty of designing the systems and the amount of data they require to learn (and, correspondingly, the quality of solutions they obtain given a fixed, limited amount of data). That being said, the diversity of possible assumptions would permit developing lifelong learning methods for various problem settings with varying degrees of data and domain-knowledge availability.

As a nascent field, lifelong compositional learning has a range of open questions, which future investigations should tackle. The following subset of such questions would likely substantially impact the broader field of AI:

- **Measuring compositionality.** Understanding precisely and quantitatively how well a solution captures the compositional nature of a problem remains a largely unsolved problem. Existing works have primarily focused on developing benchmarks that intuitively require the agent to exhibit compositional reasoning in order to solve the tasks, with a handful of exceptions that have gauged the ability of approaches to recombine existing solutions. Advancements toward quantifying compositionality would provide insight into directions to explore for creating better compositional methods.

- **Real-world applications.** While some evaluations in existing works have been inspired by realistic robotic applications, it is uncertain how well existing approaches would fare upon deployment on physical robots. More broadly, this is a challenge faced by the larger research field. Benchmark data sets and RL environments enable fast development and fair performance comparisons, both of which are useful for accelerating progress. However, *applied* research should progress. In particular, lifelong learning has so far been disconnected from real-world deployments, partly because of the artificial nature of task-based lifelong learning. Future work tackling realistic applications with embodied agents to complement fundamental lifelong learning developments would have massive impact.

- **Task-free lifelong learning.** As hinted at above, one significant step in the direction of deployed lifelong learning would be to move away from the task-based formulation. Some recent works have placed efforts in this direction, but this still remains a severely underdeveloped area. In particular, it is critical that future instantiations of the problem do not assume that individual inputs contain sufficient information to determine the agent's current objective. This assumption, which most task-agnostic works to date make, is unfortunately unrealistic. Instead, real-world (embodied) lifelong

learning would require the agent to explore and study the environment over a stream of temporally correlated inputs to discover its current objective.

- **Combining temporal and functional composition.** As discussed in Section 7, temporal and functional composition capture complementary dimensions of realistic problems. The combination of these two families of techniques has not been investigated to date. Devising mechanisms that can decompose temporally extended actions or skills into functional modules, such that these components can be reused across multiple skills, could lead to drastic data efficiency gains.

- **Flexible compositionality.** Existing works have evaluated approaches in two settings: noncompositional and compositional. Noncompositional evaluations serve to demonstrate the flexibility of the methods, while compositional evaluations serve to study the compositional properties of the algorithms. The real world is neither of these two extremes: it has a multitude of compositional properties, but many tasks require highly specialized knowledge. Devising techniques (and corresponding evaluation domains) that explicitly reason about when compositional or specialized knowledge is required would constitute another significant step toward deployed lifelong learning.

- **Other forms of composition.** While the focus of this article is to discuss the connections between lifelong learning and functional composition, it should also be clear from the surveyed works that there are numerous other forms of composition. Specifically, the RL community has developed a variety of temporal, representational, logical, and morphological views of the notion of compositionality. Each of these formulations is promising toward developing agents that accumulate knowledge and compose it in combinatorially many ways to solve a wide range of diverse tasks. Developing concrete instantiations of this intuition could potentially be highly impactful.

- **Moving beyond deep learning.** This final comment encourages future work to look beyond deep learning in the development of lifelong and compositional learners. The current trend in the field is to leverage neural network modules as the main form of compositional structures. The reason for this choice is practical: neural networks and backpropagation are today the most powerful tools available in ML, and they permit abstracting away the many nuances of statistical learning and optimization, and focus instead on the notion of knowledge compositionality. Historical evidence suggests that the tools of the future will be different from (the current version of) deep learning, and consequently future research should not focus exclusively on deep learning, and develop approaches to lifelong compositional learning that look outside of deep learning as well.

**Acknowledgments**

We thank the anonymous reviewers for their valuable comments and suggestions. J. A. Mendez is funded by an MIT-IBM Distinguished Postdoctoral Fellowship. The research presented in this article was partially supported by the DARPA Lifelong Learning Machines program under grant FA8750-18-2-0117, the DARPA SAIL-ON program under contract HR001120C0040, the DARPA ShELL program under agreement HR00112190133, and the Army Research Office under MURI grant W911NF20-1-0080. Any opinions, findings, conclusions, or recommendations expressed in this article are those of the authors and do not necessarily reflect the views of DARPA, the U.S. Army, or the United States Government.

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

# A    Appendix

Table 1: A categorization of existing works into six axes. The vast majority of work on lifelong learning has not learned explicitly compositional structures, while most efforts on compositional learning have operated in the MTL or STL settings.

| Comp. type | Lifelong? | Mechanism | Struc. given | Struc. type | Domain | References |
|---|---|---|---|---|---|---|
| None | Lifelong | Supervised | No | None | Vision | Kirkpatrick et al. (2017), Ritter et al. (2018), Zenke et al. (2017), Chaudhry et al. (2018), Serrà et al. (2018), Yoon et al. (2020), Jung et al. (2020), Cha et al. (2021), Chaudhry et al. (2020), Saha et al. (2021), Deng et al. (2021), Duncker et al. (2020), Nguyen et al. (2018), Ahn et al. (2019), Loo et al. (2021), Zhang et al. (2021), Kumar et al. (2021), Li and Hoiem (2017), Benjamin et al. (2019), Titsias et al. (2020), Pan et al. (2020), Wang et al. (2021), Chaudhry et al. (2019b), Lopez-Paz and Ranzato (2017), Chaudhry et al. (2019a), Riemer et al. (2019), Gupta et al. (2020a), Mirzadeh et al. (2021), Raghavan and Balaprakash (2021), Guo et al. (2020b), Pham et al. (2021b), Knoblauch et al. (2020), Derakhshani et al. (2021), von Oswald et al. (2020), Henning et al. (2021), Shin et al. (2017), Singh et al. (2020), Yoon et al. (2018), Hung et al. (2019), Adel et al. (2020), Lee et al. (2021a), Ke et al. (2020), Hurtado et al. (2021), Yoon et al. (2021), Mirzadeh et al. (2020), Ramasesh et al. (2021), Lee et al. (2021b), Ehret et al. (2021), Schwarz et al. (2018) |
| None | Lifelong | RL | No | None | Vision | Kirkpatrick et al. (2017), Jung et al. (2020), Wang et al. (2021), Schwarz et al. (2018), Rusu et al. (2016) |
| None | Lifelong | Supervised | No | None | Language | Del Chiaro et al. (2020), Sun et al. (2020), Ke et al. (2021), Razdaibiedina et al. (2023), Gupta et al. (2020c) |
| None | Lifelong | Unsupervised | No | None | Vision | Kumar et al. (2021), Rostami (2021), Varshney et al. (2021) |
| Functional | Lifelong | Supervised | No | Aggregation | Vision | Ramesh and Chaudhari (2022), Ruvolo and Eaton (2013) |
| None | Lifelong | Supervised | Implicitly | None | Vision | Zeno et al. (2018), Zeno et al. (2021), Kao et al. (2021), Kapoor et al. (2021), Chen et al. (2021), Yin et al. (2021), Buzzega et al. (2020), Pham et al. (2021a), Tang and Matteson (2021), von Oswald et al. (2021), Aljundi et al. (2019b), Aljundi et al. (2019a), Chrysakis and Moens (2020), Borsos et al. (2020), Jin et al. (2021), Caccia et al. (2020), Hu et al. (2019), Van de Ven et al. (2020), Aljundi et al. (2017), Jerfel et al. (2019), Lee et al. (2020), Javed and White (2019), Beaulieu et al. (2020), Banayeeanzade et al. (2021) |
| None | Lifelong | Supervised | Explicitly | None | Vision | Joseph and Balasubramanian (2020), Skorokhodov and Elhoseiny (2021) |
| None | Lifelong | Unsupervised | Implicitly | None | Vision | Egorov et al. (2021), Achille et al. (2018), Rao et al. (2019), Ayub and Wagner (2021) |
| None | Lifelong | Supervised | Implicitly | None | Language | de Masson d'Autume et al. (2019) |
| None | Lifelong | Supervised | No | None | Audio | Ehret et al. (2021) |
| Functional | MTL | Supervised | Explicitly | Graph | VQA | Andreas et al. (2016), Saqur and Narasimhan (2020), Pahuja et al. (2019), Hu et al. (2017), Johnson et al. (2017), Gupta et al. (2020b), Akula et al. (2021), D'Amario et al. (2021), Bahdanau et al. (2018) |

| Comp. type | Lifelong? | Mechanism | Struc. given | Struc. type | Domain | References |
|---|---|---|---|---|---|---|
| Functional | MTL | Supervised | Explicitly | Graph | Toy | Bošnjak et al. (2017), Zaremba et al. (2016), Cai et al. (2017), Bunel et al. (2018), Pierrot et al. (2019), Agarwala et al. (2021), Ghazi et al. (2019) |
| Functional | MTL | Supervised | Explicitly | Graph | Vision | Wu et al. (2021) |
| Functional | MTL | Supervised | Explicitly | Graph | Robotics | Xu et al. (2018) |
| Functional | MTL | Supervised | Explicitly | Graph | Language | Nye et al. (2020), Chen et al. (2020b) |
| Functional | MTL | Supervised | Implicitly | Graph | Vision | Rahaman et al. (2021), Chang et al. (2019), Rosenbaum et al. (2019) |
| Functional | MTL | Supervised | Implicitly | Graph | Language | Kirsch et al. (2018), Rosenbaum et al. (2019), Lake and Baroni (2018), Lake (2019), Ren et al. (2020), Gordon et al. (2020), Guo et al. (2020a), Liu et al. (2020), Akyürek et al. (2021), Schick et al. (2023), Keysers et al. (2020) |
| Functional | MTL | Supervised | No | Graph | Vision | Meyerson and Miikkulainen (2018), Meyerson and Miikkulainen (2019), Chen et al. (2020c), Rosenbaum et al. (2018), Rosenbaum et al. (2019) |
| Functional | MTL | Supervised | No | Graph | Language | Meyerson and Miikkulainen (2019), Chen et al. (2020c), Rosenbaum et al. (2019) |
| Functional | MTL | Supervised | No | Graph | Robotics | Alet et al. (2018), Alet et al. (2019) |
| Functional | Lifelong | Supervised | Explicitly | Chaining | Toy | Reed and de Freitas (2016) |
| Functional | Lifelong | RL | No | Chaining | Vision | Fernando et al. (2017) |
| Functional | Lifelong | Supervised | No | Chaining | Vision | Fernando et al. (2017), Li et al. (2019), Valkov et al. (2018), Veniat et al. (2021), Rajasegaran et al. (2019), Chen et al. (2020a), Qin et al. (2021) |
| Functional | Lifelong | Supervised | Explicitly | Chaining | VQA | Kim et al. (2019) |
| Functional | Lifelong | Supervised | No | Chaining | Toy | Gaunt et al. (2017) |
| Functional | Lifelong | Supervised | No | Graph | Vision | Mendez and Eaton (2021) |
| Functional | Lifelong | Supervised | Implicitly | Chaining | Vision | Ostapenko et al. (2021) |
| Functional | Lifelong | Supervised | Implicitly | Graph | Language | Li et al. (2020b) |
| Functional | MTL | Supervised | Explicitly | Aggregation | Vision | Huynh and Elhamifar (2020), Atzmon et al. (2020), Ruis et al. (2021), Du et al. (2020), Aksan et al. (2020), Arad Hudson and Zitnick (2021), Ramesh et al. (2021), Ramesh et al. (2022) |
| Functional | STL | Supervised | Implicitly | Graph | Vision | Andreas et al. (2019), Schott et al. (2022) |
| Functional | STL | Supervised | Implicitly | Graph | Language | Andreas (2019), Csordás et al. (2021) |
| Functional | STL | Supervised | Explicitly | Graph | Vision | Sylvain et al. (2020) |
| None | Lifelong | RL | Implicitly | None | Robotics | Kaplanis et al. (2019) |
| None | Lifelong | RL | Explicitly | None | Vision | Isele and Cosgun (2018) |
| None | Lifelong | RL | Implicitly | None | Vision | Rolnick et al. (2019) |
| None | Lifelong | RL | No | None | Robotics | Berseth et al. (2022), Garcia and Thomas (2019), Sutton et al. (2011), Wolczyk et al. (2021) |
| Functional | Lifelong | RL | No | Aggregation | Robotics | Bou Ammar et al. (2014), Bou Ammar et al. (2015), Mendez et al. (2020), Nagabandi et al. (2019) |
| Functional | Lifelong | RL | Explicitly | Aggregation | Robotics | Isele et al. (2016), Rostami et al. (2020) |
| Functional | MTL | RL | No | Aggregation | Robotics | Zhao et al. (2017) |
| Temporal | Lifelong | RL | No | Graph | Robotics | Xu et al. (2020), Lu et al. (2021) |
| Temporal | STL | RL | Explicitly | Graph | Toy | Sutton et al. (1999) |
| Temporal | STL | RL | Explicitly | Graph | Robotics | Lee et al. (2019) |
| Temporal | STL | RL | No | Graph | Vision | Bacon et al. (2017) |
| Temporal | MTL | RL | No | Graph | Robotics | Nam et al. (2022), Devin et al. (2019) |

| Comp. type | Lifelong? | Mechanism | Struc. given | Struc. type | Domain | References |
|---|---|---|---|---|---|---|
| Temporal | STL | RL | No | Graph | Robotics | Peng et al. (2019), Li et al. (2021b) |
| Functional | STL | RL | No | Chaining | Toy | Dayan and Hinton (1993), Dietterich (2000) |
| Functional | STL | RL | No | Chaining | Vision | Vezhnevets et al. (2017) |
| Temporal | Lifelong | RL | No | Graph | Toy | Brunskill and Li (2014) |
| Temporal | Lifelong | RL | No | Graph | Vision | Tessler et al. (2017) |
| Functional | Lifelong | RL | No | Chaining | Toy | Abel et al. (2018) |
| Functional | STL | RL | Explicitly | Aggregation | Robotics | Todorov (2009) |
| Functional | MTL | RL | Explicitly | Aggregation | Vision | Barreto et al. (2018) |
| Functional | Lifelong | RL | Explicitly | Aggregation | Toy | Alver and Precup (2022) |
| Functional | MTL | RL | Explicitly | Aggregation | Robotics | Haarnoja et al. (2018), Cheng et al. (2021), Li et al. (2021a), Bylard et al. (2021), Jothimurugan et al. (2019) |
| Functional | MTL | RL | Explicitly | Aggregation | Toy | Van Niekerk et al. (2019), Colas et al. (2019), Nangue Tasse et al. (2020) |
| Functional | Lifelong | RL | No | Aggregation | Toy | Nangue Tasse et al. (2022) |
| Temporal | MTL | RL | Explicitly | Graph | Robotics | Jothimurugan et al. (2021), Qiu and Zhu (2022) |
| Functional | STL | RL | Explicitly | Aggregation | Vision | Van Seijen et al. (2017) |
| Functional | STL | RL | No | Aggregation | Vision | Lin et al. (2019), Lin et al. (2020), Mu et al. (2020) |
| Functional | MTL | RL | No | Chaining | Robotics | Devin et al. (2017), Yang et al. (2020), Mendez et al. (2022a) |
| Functional | STL | RL | No | Graph | Vision | Mittal et al. (2020), Goyal et al. (2021), Goyal et al. (2022) |
| Functional | STL | Supervised | No | Graph | Language | Mittal et al. (2020) |
| Functional | STL | Supervised | No | Graph | VQA | Goyal et al. (2022) |
| Functional | Lifelong | RL | No | Chaining | Robotics | Mendez et al. (2022b) |
| Functional | MTL | RL | No | Graph | Robotics | Pathak et al. (2019), Huang et al. (2020) |
| Functional | STL | RL | No | Aggregation | Robotics | Li et al. (2020a) |
| Functional | MTL | RL | No | Graph | Toy | Chang et al. (2021) |
| None | STL | Supervised | No | None | Vision | Deng et al. (2009) |
| None | STL | RL | No | None | Vision | Bellemare et al. (2013), Vinyals et al. (2017) |
| None | STL | RL | No | None | Robotics | Brockman et al. (2016), Tunyasuvunakool et al. (2020) |
| None | MTL | RL | No | None | Robotics | Henderson et al. (2017), Yu et al. (2019), James et al. (2020) |
| None | MTL | RL | Explicitly | None | Toy | Chevalier-Boisvert et al. (2019) |
| None | Lifelong | RL | No | None | Toy | Nekoei et al. (2021) |
| Functional | Lifelong | Supervised | Explicitly | Graph | Toy | Sinha et al. (2020) |
| Functional | MTL | RL | Explicitly | Chaining | Robotics | Mendez et al. (2022a) |
| Temporal | MTL | RL | Explicitly | Graph | Toy | Gur et al. (2021) |
| Functional | MTL | RL | Implicitly | Graph | Robotics | Ahmed et al. (2021) |

