# OpenReview forum: "How to Reuse and Compose Knowledge for a Lifetime of Tasks: A Survey on Continual Learning and Functional Composition"
_TMLR — Accepted by TMLR_

### Review · Reviewer_ebXm · 2023-02-28

**Summary Of Contributions:**

This paper’s contributions are as follows:
1. A survey of lifelong learning, including a categorization of the problem statements within lifelong learning as well as a categorization of potential approaches to these in language, vision, and robotics domains.
2. A survey of compositional learning, including a delineation between temporal and functional composition.
3. A call for research to investigate the use of compositional methods in lifelong learning.


**Audience:**

Yes

**Broader Impact Concerns:**

Compositional and lifelong learning approaches will output different agents than unstructured, iid learning. How do they improve or worsen issues of bias and fairness?


**Claims And Evidence:**

Yes

**Requested Changes:**


I suggest adding some discussion to the points above.

——————————

There are a few statements in the paper that are incorrect or insufficiently supported.

 “AI systems in the real world will not have access to batches of simultaneous tasks, but instead will face them in sequence in a lifelong setting”.
-> An agent in the real world will likely be simultaneously parsing sound into text, predicting future video input, detecting object, estimating human poses, etc. There is no reason it would have to learn only one task at a time in sequence and then never see data for that task again.

“In order for knowledge to be maximally reusable, it must capture a self-contained unit that can be composed with similar pieces of knowledge”
-> This statement can be either viewed as a strong claim that needs significantly more evidence to say, or as vague philosophizing that doesn’t really say anything at all.

“A service robot that has learned to both search-and-retrieve objets and navigate across a university building should be able to quickly learn to put these together to deliver a stapler to Rosie’s office. Instead, typical methods […]”
-> This is how most hierarchical policy learning works already, which is common and old enough to be considered a typical method.

——————————

In Figure 4, the supervised vs unsupervised delineation may not be necessary. Unsupervised learning is usually algorithmically equivalent to supervised, it just denotes whether the supervision is from a human or from a non-human (such as physics, time, etc). At the same time, there are only 2 papers that fall in the unsupervised bucket. This figure could also be transposed to have application domain as rows, and supervised vs RL as the marker-fill, which would be easier to read.



**Strengths And Weaknesses:**

The paper presents a quite thorough analysis of the problem settings within the scope of lifelong and compositional ML. Many relevant prior works are discussed and categorized across multiple axes. The cross section of lifelong learning and compositionality is an interesting area to explore.

A strength of the paper is in its detailed categorization of prior works in this area, in terms of problem setting, type of supervision, algorithmic approach, and architectural approach. The difference between temporal and functional composition is especially good to have written down as these concepts are often discussed interchangeably.

The figures in the paper are well designed (except for Figure 4)

The paper concludes with a nice list of further work to be done in the field.

——

One weakness of the survey is a lack of criticism of the standard lifelong learning (LL) problem statement, especially in regards to the feasibility of learning compositional agents in an LL setting. As described in the paper, the standard LL problem setting is to maximize performance on all tasks while only being able to train on each task once, in sequence.
The artificialness of this definition is discussed in the final paragraph before 2.2.1, which explains exactly why the standard LL problem doesn’t actually address a realistic setting, as it would be impossible to continually learn new tasks, and never forget ones that are never seen again, with a finite memory. However, the standard LL problem statement is particularly adversarial to compositional methods, as learning how to best decompose multiple tasks into reusable components is easiest when you can train on multiple tasks simultaneously, even if the overall distribution across tasks changes over time consistent with the broader LL motivation.

The work would also benefit from a more specific discussion of what is generally assumed when LL or compositional approaches are used. For example, a task-agnostic setting where always the same input is mapped to different outputs depending on the (hidden) task is an arbitrarily difficult LL problem. Past tasks can only accelerate future learning if they are in some way related. Compositional structures can lead to worse performance if they don’t match the underlying structure of the environment, as they are a form of bias onto the final model.

The paper successfully describes the width of prior work across	LL and compositional learning, but it could include a bit more discussion on which LL methods are most promising for composition, and vice versa.

An early paper on functional compositional learning in robotics that the authors may want to include: “Learning modular neural network policies for multi-task and multi-robot transfer” (Devin 2017)

---

> ### Author Response · Authors · 2023-04-26
> **Response to review from ebXm**
>
> 1. **Lack of criticism to standard lifelong setting:** While we agree with the reviwer that the prevalent lifelong learning paradigms studied in the literature are to varying extents artificial, we strongly believe that the strictly sequential nature of the problem is not a weakness of the formulation. Learning compositional structures is certainly complicated by this strictly sequential setting, but that difficulty lies at the core of the lifelong compositional learning problem. For example, a home assistant robot may be deployed for a long time before it ever faces a task that requires opening a door. It is unreasonable to expect the robot to learn this new "component" by re-learning how to solve all previous tasks that require no door opening. We view this type of sequential decomposition as precisely the crux of the matter. Of course, specific applications of the problem may relax this constraint in a way that suits the concrete domain (e.g., maybe robots can train some initial decomposition in a set of pre-training tasks before being deployed for lifelong training).
>
> 2. **Discussion of lifelong and compositional assumptions:**
>   * The approaches of various forms of lifelong learning methods are delineated in Section 2.1. The point the reviewer raises, regarding the arbitrary difficulty of mapping equivalent inputs to different outputs given hidden task information, is absolutely on point, and yet the literature mostly fails to discuss this explicitly. Our submission explicitly addresses this point in Section 4.2.5.
>   * The various compositional assumptions are also discussed extensively in Section 2.2. We will expand our discussion to clarify that this form of bias may be detrimental if the _problem_ doesn't match the _method's_ assumptions (as the reviewer points out).
>
> 3. **Discussion about promising lifelong approaches for composition and vice versa:** This survey seeks to make the case that explicitly modular and compositional models are promising candidates for addressing the lifelong learning problem. While we largely refrain from making value judgments in order to offer a balanced review of the state of the field, we do state this claim in Sections 5 and 7 alongside the respective methods, and reiterate it more explicitly in Section 8.
>
> 4. **Cite to Devin et al. 2017:** Our original submission discussed Devin et al.'s work as an initial method in functionally compositional RL (page 26, bottom half).
>
> 5. **Suggested edits/revisions:**
>   * "AI systems in the real world..." -- This is a good point. We meant to convey that the agent will be solving a single objective at a time (e.g., set the table, load the dishwasher...). But, as the reviewer rightly points out, there is no need for these higher-level objectives to match one-to-one with the agent's learning tasks. We will address this case in the revised manuscript.
>   * "In order for knowledge..." -- As stated in the clause immediately preceding the quoted text, this is merely an intuition and not a formal statement.
>   * "A service robot that has..." -- While indeed hierarchical RL approaches have existed for a while, 1) they are by no means the "typical" method used today, and 2) they are exceedingly rarely used in the lifelong setting, which is what the paragraph in question refers to.
>
> 6. **Figure 4:** We took the reviewer's suggestion to unify the supervised and unsupervised rows in Figure 4, and have drafted a revision. We do agree it makes the illustration much clearer (thanks!), and will include it in the revised manuscript. We will still maintain the distinction in the Appendix for completeness and precision.

---

### Review · Reviewer_4YsP · 2023-03-31

**Summary Of Contributions:**

This paper is a survey of recent works (focussing mainly on the most recent five years) of life-long learning and continual learning, including all forms of machine learning where these are relevant - (un)supervised and reinforcement learning. The paper provides a careful categorization of these methods - in terms of the framing such as task-based and task agnostic, in terms of algortihmic and data approach, and to some extent (esp. in the RL section) in terms of how an application domain becomes decomposed, etc. To the extent that this is mainly a survey of past work, the innovation is in the interpretation of the methods in order to extract the shared structures.

**Audience:**

Yes

**Broader Impact Concerns:**

This is a survey paper on existing work, so as such this paper does not raise any new concerns about broader impact. The paper does a nice job of synthesizing the work from the past five years and of explaining how different strands of work relate to each other.

**Claims And Evidence:**

Yes

**Requested Changes:**

Having said that this is a well written and clear paper that makes a timely contribution, the survey would be stronger if the authors find ways to address weaknesses outlined in the previous section. I will not reiterate the list here, but each weakness might require further thought about how best to organise and explain the works.

**Strengths And Weaknesses:**

The primary strength of this paper is that it is suitably comprehensive, as a survey should be, in covering the the relevant literature. I like the following elements:
- useful categorization in terms of types of machine learning methods, types of algorithmic approach and data assumptions, task decomposition, etc.
- This is chiefly but not exclusively in terms of the six axes of variation, which are well chosen
- the writing is clear and the use of visualizations is helpful in bringing out and supporting the structures being addressed
- to the best of my knowledge the coverage of the papers is also adequate

The primary weaknesses of this paper are as follows:

- The different sections addressing lifelong and compositional learning, supervised and reinforcement learning, etc. are indeed distinct from each other. While within each section, there is an appropriate form of analysis and categorization, I do not get a higher level understanding of principles that cut across the top level categories. So, for instance, if one asks how the compositional templates (such as chaining or aggregation, or even graphs) relate to, say, similar questions in the RL section then the paper does not quite explain this. Of course, there are indeed connections, e.g. there is graph structuring in both, but both the structuring of the sections and the ways of writing do not expose any such connections.
This is important because the introduction explicitly starts by making the case for how the many different types of learning - lifelong, continual and compositional, multi-task, etc. are all essentially related and should be studied as such

- A second major point is that the focus on just papers from 2017 seems a bit odd for subject matter that has indeed been thought about since quite a bit earlier. To be fair to the authors, they do mention key early works (such as Thrun) and the recent explosion of ML literature would make it hard to be entirely comprehensive over a much longer period. However, this has the effect that many other forms of decomposition of tasks and analysis of stucture are left out unless featured in the deep learning literature of the recent couple of years

- There is a distinct lack of consideration of methods that approach representations in other ways, e.g. there is not much discussion of things like options or sub-task discovery in the RL setting, neuro-symbolic representations and algorithms that make use of symbolic structuring or reasoning in any way, etc.

- There is a distinct lack of specific domain examples - from vision, language and so on, and much of the discussion is in terms of machine learning models alone. This makes the paper rather dry. That said, the examples in the RL section are quite useful (e.g. fig 8) and more such examples in the earlier sections would have been very welcome

---

> ### Author Response · Authors · 2023-04-26
> **Response to review by 4YsP**
>
> 1. **Sections are distinct/disjoint:** This is an excellent point to raise. With such a broad-reaching survey, there were certainly major efforts employed towards structuring the article in a way the exposed the connections across "top level categories". In particular, the main thread that we used to examine those connections was the six-axis categorization of Figure 4. Note that we applied the categorization to _all_ the major references, as summarized in the Appendix. We did attempt to highlight these interconnections in text. For example, we mention how several RL approaches relate to the compositional structures in terms of aggregation/chaining/graph. However, we do recognize that these cross-cutting discussions could be better highlighted and explained, which we will address in the revised manuscript per your suggestion.
>
> 2. **Chronological scope:** As stated in Section 1.1, our survey indeed focuses on works published on 2017 or later. While there is certainly relevant earlier literature, attempting to cover _all_ relevant literature from the past would unfortunately be intractable. In particular, earlier _lifelong learning_ literature is quite sparse, and we did include the most relevant references we know of from those earlier years, including Sebastian Thrun's seminal work (as the reviewer rightly points out) and Ruvolo et al.'s ELLA paper. Earlier _compositional learning_ literature is on the contrary quite substantial and broad. Consequently, we focused on the compositional works that have been adapted to the deep learning landscape. While there is of course opportunity to investigate alternative approaches that _haven't_ been adapted to deep learning, we contend that it falls outside of the scope of this survey to find those connections, which would each likely constitute an algorithmic paper on its own right. That being said, if the reviewer has specific suggestions for literature that we have missed that would aid in providing context or insight in the scope of the submission, we would be more than glad to incorporate it.
>
> 3. **Alternative compositional representations:**
>   - Options/sub-task decomposition in RL. Please note that the focus of our survey is on _functional_ composition, and as such option discovery is only described for context and _not as a standalone survey of hierarchical RL_. We had stated this briefly towards the bottom of page 24 ("With this in mind, note that the discussion of works on skill discovery, which is a vast literature on its own, is by no means comprehensive") and have will expand the disclaimer and point to relevant surveys in that space.
>   - Neuro-symbolic / reasoning approaches. This is an excellent point. Our original submission included multiple references from symbolic programming and other forms of reasoning-based approaches, but we had not explicitly stated this distinction. We will add a separate paragraph highlighting this point.
> 4. **Examples from vision, language, and others:** Thank you for pointing this out. We will add examples in earlier sections, starting from the introduction, that we can carry throughout in similar vein to the RL section.

---

### Review · Reviewer_zbLo · 2023-04-11

**Summary Of Contributions:**

The paper investigates two important and practical problems: lifelong learning (LL) and compositional learning (CL). It fills an important gap by combining and unifying both approaches in a single survey. It clearly divides each problem into semantically meaningful categories that help position each problem and approach. It gives a broad coverage of LL and CL approaches under a variety of settings in a single accessible survey. Carefully designed figures at each section give a very useful summary and help visualize different approaches.

**Audience:**

Yes

**Broader Impact Concerns:**

This is a survey paper with not ethical concerns.

**Claims And Evidence:**

Yes

**Requested Changes:**

In addition to my concerns in the weaknesses above,

- Please clarify that $F^{\hat{t}}$ is the fisher information matrix.

- Please add respective section names to Figure-5 to easily find the related details.

**Strengths And Weaknesses:**

**Strengths**

- The paper is easy to follow with a good a summary of approaches for LL and CL as well as their intersection. It will be useful for the research community to get an overview of both topics.

- Figures/table were carefully designed to get a high level overview of each section. For example, Figure 6 describes compositional learning from input/output and latent/observed information perspectives. Table-1 in the appendix gives a nice view of all the related work under main verticals such as type of compositionality, labeling mechanism, etc. that the paper has been based on from the beginning.

- The paper nicely presents a hierarchical clustering of concepts within LL and CL including their intersection. Both LL and CL are explained under several broad categories that capture many of the practical tasks; there are 3 categories for LL: (i) Task-, (ii) Domain-, and (iii) Class-incremental learning; there are 2 categories for CL: (i) models and (ii) problems. Lifelong compositional learning is further divided into 6 additional categories to capture a more unified view.

- Dividing the survey into supervised learning (SL) and reinforcement learning (RL) made the paper much easier to follow. Using SL as the prior was also helpful to get better understanding of RL approaches.

**Weaknesses**

- The Section 2.1 emphasizes MTL much more than the typical streaming LL problem -- streaming tasks. It would help clarify the differences in training objectives by clearly stating what information is accessible and which period of time is optimized. In addition, writing separate objectives for training and evaluation where it is appropriate could also help.

- Could you please discuss if two different views of compositionality, function composition and graph-based composition, are equivalent? Based on Figure-3 (c), functions are placed after every rectangular node which suggests that it doesn't correspond to a chain of functional composition. Traversing left-to-right by grouping functions in the graph might give you a functional composition but not clear if this generalizes beyond a DAG-based structure.

- Presenting a more formal definition that unify core approaches in LL and CL would really help understand the paper better. For example, a unified form of the *regularization* objective, such as KL-control or parameter masking, integrating *replay* buffer with a sampling mechanism, or selecting different modules in *compositional learning*.

- You mention forward and backward transfer in earlier sections but without any pointers later. I think these are two really important topics to understand how different methods are helping improve which timespan of the streaming tasks. Please mention these where appropriate.

- The paper is missing a major related work -- pretrained language or image models in lifelong and compositional learning. While I found one mention of BERT, pretrained models are mainly missing from the survey. Given that they are the main paradigm nowadays, the survey is missing the most up-to-date research stream with the highest practical importance. Several categories that should be included are: few-shot prompting, parameter efficient tuning (p-tuning) in LL [1], and tool use for compositionality [2].

- Time and memory complexities are missing. I think these are important to understand the practical usage and further gain insight into drawbacks of different methods.

- While there is a section devoted to benchmarks in RL, these are largely missing in the SL setting. It is not clear which datasets/benchmarks are studied in the SL and what are some of the remaining challenges in these benchmarks.

[1] Progressive Prompts: Continual Learning for Language Models. Anastasia Razdaibiedina, Yuning Mao, Rui Hou, Madian Khabsa, Mike Lewis, Amjad Almahairi.

[2] Toolformer: Language Models Can Teach Themselves to Use Tools. Timo Schick, Jane Dwivedi-Yu, Roberto Dessì, Roberta Raileanu, Maria Lomeli, Luke Zettlemoyer, Nicola Cancedda, Thomas Scialom.

---

> ### Author Response · Authors · 2023-04-26
> **Response to review from zbLo**
>
> 1. **Multitask vs lifelong learning:** In the formulation we adopt in our work, the lifelong and the multitask objectives are identical: to optimize performance across all tasks. This matches the formulation made (either explicitly or implicitly) in the large majority of existing lifelong learning works. The difference lies in that MTL agents receive data from all tasks simultaneously, while lifelong learners receive data from tasks consecutively, in sequence. This latter point implies that a lifelong learning agent must optimize the performance of all tasks seen so far at any point in time. This is specified in the definition of the objective _z_ (pg 5) in the task-incremental setting. The _training_ objective is entirely dependent on the learning algorithm, but the goal of a lifelong algorithm is to approximate the true MTL objective as computationally efficiently as possible.
>
> 2. **Graph vs functional composition:** Yes, graph and function composition are equivalent. In particular, we use a graph to formalize how functions are composed. The graph formalism captures more complex expressions than a "chain of functional composition", since (among others):
>  * nodes with in-degrees > 1 can consume the outputs of various previous nodes as inputs,
>  * nodes with out-degrees > 1 can produce inputs to multiple subsequent nodes,
>  * in principle, cyclical graphs represent a form of functional recursion, which can also be viewed as composition
>
> 3. **Formal, unified definitions of compositional and lifelong methods:** This is a great suggestion, and we will incorporate it into the survey via a direct combination of the mathematical formulations already specified in Sections 2.1 and 2.2. While we had made our best effort to provide these unified definitions _in text_ in these same sections, we agree that providing a  _mathematical_ version of these commonalities in a new Section 2.3 will improve the quality of the manuscript.
>
> 4. **Forward and backward transfer:** We deliberately chose to focus our survey on the broader notion of _reusable knowledge_, and will clarify in Section 1 how forward and backward transfer relate to the reusability of knowledge. That being said, given that the majority of the lifelong learning community is accustomed to discussing works in terms of forward/backward transfer, we will revise our manuscript to point to these concepts where appropriate, particularly in Section 4.
>
> 5. **Pre-trained models:** The paradigm of pre-trained models is certainly a promising one towards developing lifelong/compositional methods. However, it is our perspective that such models fall outside of the scope of our survey, for the reasons outlined below:
>   * In the context of compositionality, pre-trained models have the potential to exhibit compositional properties extracted from the large-scale data. However, they are not themselves explicitly compositional or modular, therefore making it unclear how to add/remove components over time, improve existing components, adapt them to shifting distributions... We briefly discuss this in our submission in Section 5.3, with references to the works of Ramesh et al. (2021, 2022).
>   * In the context of continual learning, the literature has not studied the use of pre-trained models broadly enough to understand its implications in depth. Our survey describes the work Ke et al. (2021), which is among the very few published works that have considered the paradigm of pre-trained models in the context of continual learning.
>   * Regarding the specific references suggested by the reviwer, they were both made public _after_ the submission of our current manuscript (so, of course, it would not have been possible for us to include them). However, we would be pleased to include it in a revision. More broadly, this is a line of work that is receiving tremendous attention in the very recent past, and we would look forward to future stand-alone surveys that scrutinize it once the field is better understood.

---

> ### Author Response · Authors · 2023-04-26
> **Response to review from zbLo (Part 2)**
>
> 6. **Time and memory complexities:** Unfortunately, tracking down or deriving the computational and memory complexities of all existing works is intractable, especially given that most works do not disclose these costs for their proposed methods. Roughly, regularization methods are more time and memory efficient than replay methods, and generative replay methods are more memory efficient than replay method but less computationally efficient. However, the variance within each category is quite large, and so, while we are a bit skeptical that adding these broad comparisons would be especially valuable, we can include them in the revision if you still think it important. We would welcome your insight on this.
>
> 7. **Supervised learning benchmarks:** We gave this point a considerable amount of thought prior to submitting our survey, and made the decision to omit supervised learning benchmarks from the survey for the following reasons:
>   * There is no unified definition of benchmark problems in continual learning, and it is quite common for each published work to introduce its own variant of a continual learning problem
>   * Most supervised learning benchmarks for continual learning (e.g., permuted MNIST, split CIFAR...) offer no insight into the continual learning problem, since the relations between tasks are not well understood. Therefore, this portion of the survey would likely turn out to be a simple list of existing benchmarks, without much value added in terms of understanding the current state of the field.
>   * As explained in Section 2.1.1, there are existing surveys on continual learning that have covered the most common benchmarks and evaluation settings. We do not believe that our survey would create additional value in that regard.
> Consequently, for the time being we will continue to omit coverage of supervised learning benchmarks. We are happy to discuss this point further and would be willing to reconsider if it becomes clear that the survey would be stronger by the inclusion of this point.
>
> 8. **Other Requested Changes** Thank you for pointing out these minor edits; we will make them.

---

### Author Response · Authors · 2023-04-26
**Thank you for the thought-provoking reviews**

Thank you very much to all reviewers for your comments on our manuscript. We really appreciate the thoughtfulness and thoroughness of your feedback and suggestions. We are eager to engage you in conversation regarding your suggestions, so we hope that you'll follow up to our responses below.

---

### Author Response · Authors · 2023-05-11
**Added revised version**

We have updated a revised draft of our survey incorporating the reviewers' suggestions and addressing their concerns. We would like to once again thank the reviewers for their comments, which we believe have strengthened our submission. Below, we summarize the main changes that we have made to our manuscript. Additions in the PDF are highlighted in **BLUE**.

1. **Formal, unified definitions of compositional and lifelong methods:** Equations 1 (pg 5), 2 (pg 12), 3 (pg 13), and 4 (pg 13) now provide mathematical formulations that capture the objective functions for the general lifelong setting, the parameter regularization approach, the functional regularization approach, and the replay approach. The first paragraph of pg 19 now formally describes how modular networks relate to the graph formalism of Section 2.2.
2. **Forward and backward transfer:** Section 1 discusses how forward and backward transfer relate to the reusability of knowledge in general terms. Our revised draft explicitly describes forward and backward transfer for regularization (pg 13 bottom half), replay (pg 14 bottom half), and capacity expansion (pg 15 center) methods.
3. **Sections were distinct/disjoint:** We have highlighted the connections across sections more explicitly in the following places:
    * Pg 23 third paragraph relates lifelong compositional methods in the supervised setting to the lifelong learning categorization of Section 4
    * The various paragraphs on lifelong RL in Section 6 now refer back to the equivalent lifelong supervised methods of Section 4.
    * Section 7 more explicitly calls out the statements that relate compositional RL methods to the aggregated/chaining/graph axis of our categorization in Figure 4.
4. **Alternative compositional representations:**
    * Options/sub-task decomposition in RL. We have further clarified why we choose to focus on functional instead of termporal composition in RL (pg 28 top) and have referred readers to relevant surveys in hierarchical RL (pg 28 center).
    * Neuro-symbolic / reasoning approaches: we have added a paragraph stating which class of methods from our list of references can be considered symbolic (pg 22 top).
5. **Examples:** We have expanded our example from the vision domain from Section 2.2 to Section 5.1 (pg 19 center and Figure 8).
6. **Discussion of lifelong and compositional assumptions:** We have discussed the potential negative impact of using compositional methods in a new Section 5.5.
7. **Figures**: We have revised Figures 4 and 5 as suggested by the reviewers.

---

### Decision · Action_Editors · 2023-05-18

**Recommendation:** Accept as is

**Comment:**

zbLo: "The paper serves as a good survey of more traditional approaches but lack more recent LLMs and image models. I worry that it would make the survey less relevant very quickly." "It fills an important gap by combining and unifying both approaches in a single survey"

ebXm: "this paper presents a thorough analysis of the problem settings within the scope of lifelong and compositional ML. As a survey paper, it can serve as a good foundation for new researchers to learn about the topic and as a source of areas that need further study."

4YsP: "The primary strength of this paper is that it is suitably comprehensive, as a survey should be, in covering the the relevant literature"

**Audience:**

The research communities interested in the lifelong / continual learning and compositional learning would find the survey helpful. In particular, the survey nicely structures the common problems and methods unifying the two fields.

**Claims And Evidence:**

The paper surveys recent literature on two related fields, lifelong and compositional learning. The reviewers agree, and I concur, that the survey is timely and well-positioned. The consensus is that survey would be beneficial and helpful to the community. The authors addressed the reviewers suggestions and improved the quality of the paper.

Since this is a rapidly emerging field, the camera-ready version should include some the more recent works such as [1] and [2], as suggested by the reviewer zbLo.